# Structural basis of template strand deoxyuridine promoter recognition by a viral RNA polymerase

Alec Fraser[1,7], Maria L. Sokolova [1,2,7✉], Arina V. Drobysheva[2], Julia V. Gordeeva[2], Sergei Borukhov [3], John Jumper [4], Konstantin V. Severinov [2,5,6✉] & Petr G. Leiman [1✉]

Recognition of promoters in bacterial RNA polymerases (RNAPs) is controlled by sigma subunits. The key sequence motif recognized by the sigma, the −10 promoter element, is located in the non-template strand of the double-stranded DNA molecule ~10 nucleotides upstream of the transcription start site. Here, we explain the mechanism by which the phage AR9 non-virion RNAP (nvRNAP), a bacterial RNAP homolog, recognizes the −10 element of its deoxyuridine-containing promoter in the template strand. The AR9 sigma-like subunit, the nvRNAP enzyme core, and the template strand together form two nucleotide base-accepting pockets whose shapes dictate the requirement for the conserved deoxyuridines. A single amino acid substitution in the AR9 sigma-like subunit allows one of these pockets to accept a thymine thus expanding the promoter consensus. Our work demonstrates the extent to which viruses can evolve host-derived multisubunit enzymes to make transcription of their own genes independent of the host.

[1] Department of Biochemistry and Molecular Biology, Sealy Center for Structural Biology and Molecular Biophysics, University of Texas Medical Branch, Galveston, TX 77555-0647, USA. [2] Center of Life Sciences, Skolkovo Institute of Science and Technology, Moscow 121205, Russia. [3] Department of Cell Biology and Neuroscience, Rowan University School of Osteopathic Medicine at Stratford, Stratford, NJ 08084-1489, USA. [4] DeepMind Technologies Limited, London, UK. [5] Institute of Molecular Genetics, Russian Academy of Sciences, Moscow 123182, Russia. [6] Waksman Institute for Microbiology, Rutgers, The State University of New Jersey, Piscataway, NJ 08854, USA. [7] These authors contributed equally: Alec Fraser, Maria L. Sokolova. ✉email: maria.sokolova@skolkovotech.ru; severik@waksman.rutgers.edu; pgleiman@utmb.edu

B*acillus subtilis* "jumbo" bacteriophage AR9 encodes two distinct multisubunit DNA-dependent RNA polymerases (RNAPs), allowing for the transcription of viral genes to proceed independently of the host RNAP[1–4]. The virion-packaged RNAP (vRNAP) is delivered into the host cell together with phage DNA at the onset of infection. The vRNAP then transcribes early phage genes, including those of the second, non-virion RNAP (nvRNAP). The nvRNAP transcribes late genes, including those coding for the vRNAP, which are packaged into progeny phage particles together with phage DNA.

This strategy is used by a large number of jumbo phages and, in addition to AR9, has been studied in *Pseudomonas aeruginosa* phage phiKZ[5,6]. Sequence-wise, nvRNAPs, and vRNAPs of jumbo phages can be as diverse from each other as they are from their most probable ancestors—the bacterial RNAP[7]. This evolutionary relationship has been recently confirmed by the atomic structure of the phiKZ nvRNAP[8]. However, none of the structural aspects of the nvRNAP or vRNAP function or regulation (e.g., initiation, elongation, or termination of transcription) have been defined thus far. The mechanism of promoter recognition is particularly interesting in phages that use modified or alternative bases in their genomic DNA. Notably, unlike that of phiKZ, the double-stranded (ds) genomic DNA of AR9 contains uracils instead of thymines throughout[1,2].

The catalytically active AR9 nvRNAP core enzyme consists of four proteins that, when pairwise concatenated, show about 20% sequence identity and cover the entire lengths of the universally conserved β and β′ subunits of bacterial RNAPs (Fig. 1a)[3]. Promoter-specific transcription is performed by a five-subunit holoenzyme that, in addition to the nvRNAP core, contains the product of AR9 gene *226* (gp226)[3]. Gp226 shows no discernible sequence similarity to bacterial RNAP promoter specificity σ subunits or any known transcription factor. Close orthologs of gp226 are found in the genomes of other jumbo phages that have been demonstrated or are presumed to contain uracil in their genomic DNA[9,10].

Unlike bacterial RNAPs[11], the AR9 nvRNAP holoenzyme recognizes promoters in the template strand of dsDNA and is capable of promoter-specific transcription initiation on single-stranded (ss) DNA[3]. The AR9 nvRNAP template-strand promoter consensus $3'-^{-11}UUGU^{-8}-N_6-AU^{+1}-5'$ (where N is any nucleotide and the transcription start site (TSS) coordinate is +1) contains a four base-long motif centered about 10 nucleotides upstream of the TSS and a two-base motif at the TSS (Fig. 1b). Promoters with thymines at the −11th and −10th positions are inactive, suggesting that the C5 position of the uracil's pyrimidine ring, which carries a methyl group in the thymine, plays a critical role in promoter recognition (Fig. 1c, Supplementary Fig. 1, Supplementary Data 1). Despite possessing a short promoter consensus element, the AR9 nvRNAP holoenzyme protects an extensive region of DNA flanking the TSS (position −35 to +20 in the template strand and positions −29 to +17 in the non-template strand) from DNase I attack, implying additional contacts with DNA[3].

To understand the uracil-specific, template strand-dependent promoter recognition mechanism of the AR9 nvRNAP, we determined the structure of this enzyme by X-ray crystallography and cryo-electron microscopy (cryo-EM) in three states—the core, holoenzyme, and holoenzyme in complex with a 3′-overhang dsDNA oligonucleotide that mimicked the downstream half of the transcription bubble (historically called a "forked" or "fork" template)—and complemented this structural information by discriminative in vitro transcription assays. In its ss part, the forked oligonucleotide contained the P077 promoter of the AR9 late gene *076,* which encodes a highly expressed virion protein of unknown function[2].

## Results and discussion

**Brief description of structural data.** Supplementary Table 1 lists structural datasets, their features, and their use in figures and tables in this manuscript.

We have crystallized the AR9 nvRNAP core (Fig. 1a) in two different crystal forms that had a "Standard" and "Large" unit cell containing two and eight nvRNAP core molecules in the asymmetric unit, respectively. The Standard and Large unit cell crystals diffracted X-rays to 3.30 Å and 3.79 Å resolutions, respectively (Supplementary Table 2). The electron density in all datasets was poor and displayed a large amount of disorder, which complicated the process of atomic model building. The holoenzyme (Fig. 1a) failed to produce crystals, so its structure was determined by cryo-EM to a resolution of 4.4 Å (Supplementary Tables 3, 4).

The structure of the promoter complex has been determined by both X-ray crystallography and cryo-EM to resolutions of 3.38 and 3.80 Å, respectively (Supplementary Tables 2, 3, 4). Two different crystal forms of the promoter complex were obtained. Both forms displayed a common crystal packing property that is described and illustrated in detail below. The conformations of the enzyme in both crystal forms were similar but the quality of the DNA electron density differed. The promoter complex dataset with a higher fraction of ordered DNA was used for analysis. In the structure of the same complex analyzed by cryo-EM, only three nucleotides of the promoter $(3'-^{-11}UGU^{-9}-5')$ were sufficiently ordered for model building. The cryo-EM- and X-ray-derived structures of the holoenzyme in the promoter complex were very similar and could be superimposed with a root mean square deviation (RMSD) of 1.84 Å between 2563 and 97% of all Cα atoms comprising the holoenzyme.

Considering that the X-ray structure of the AR9 nvRNAP promoter complex is more complete, has higher resolution, and is very similar overall to the cryo-EM-derived promoter complex structure, the former is used below for the description of various structural features of the enzyme such as the conformation of the active site and the folds of various domains (Supplementary Table 1). Where required, we will distinguish the X-ray and cryo-EM-derived structures of the AR9 nvRNAP promoter complex by referring them as AR9 nvRNAP-Pro-Xray and AR9 nvRNAP-Pro-cryoEM, respectively (Supplementary Table 1). The X-ray structure of the core will be referred to as AR9 nvRNAP-core-Xray (Supplementary Table 1). The cryo-EM map and the associated atomic model of the template-free holoenzyme is referred to as AR9 nvRNAP-holo-cryoEM (Supplementary Table 1).

**Structural comparison of the AR9 nvRNAP with bacterial RNAPs.** In the most populous class of the cryo-EM reconstruction and in both available crystal forms of the AR9 nvRNAP promoter complex (i.e., AR9 nvRNAP-Pro-cryoEM and AR9 nvRNAP-Pro-Xray), the enzyme bound not one but two copies of the P077 promoter-containing forked oligonucleotide—the downstream copy (as designed) and (fortuitously) the upstream one—resulting in a superstructure that resembles the complete transcription bubble found in open complexes formed by other RNAPs (Fig. 2a, b). Moreover, *in crystallo* the nvRNAP molecules and the oligonucleotides formed a train in which the upstream and downstream oligonucleotides belonging to two neighboring unit cells pi-stacked and formed a continuous double helix (Supplementary Fig. 2).

The overall structure of the AR9 nvRNAP is a trimmed-down version of a bacterial crab claw-shaped RNAP (Figs. 2a, 3). No domain compensates for the absence of α and ω subunits that are present in all bacterial, eukaryotic, and archaeal enzymes[12,13]. As

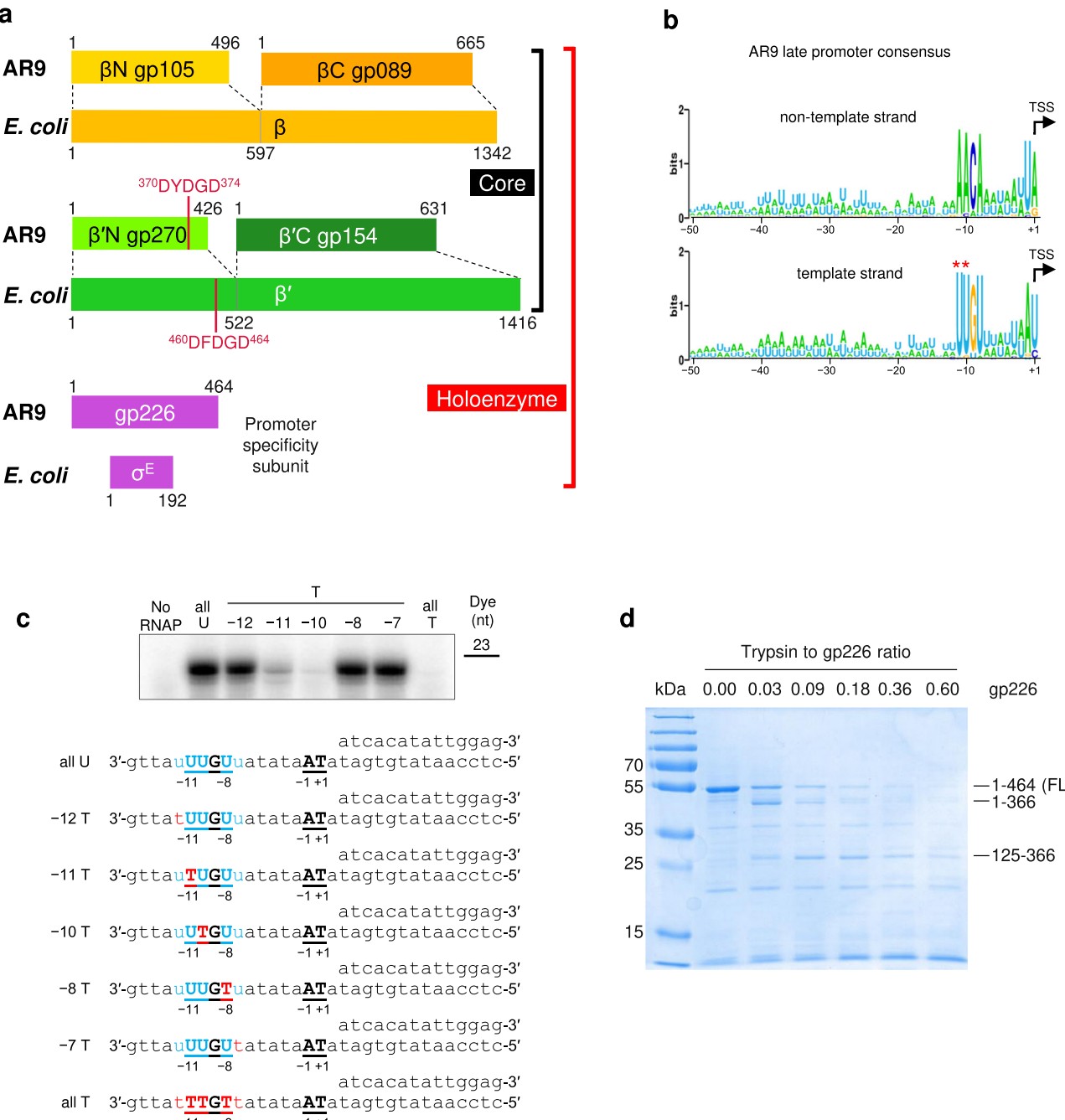

**Fig. 1 Organization and promoter consensus of the AR9 nvRNAP. a** Organization of the catalytically active core and promoter initiation-competent holoenzyme of the AR9 nvRNAP. A pair of genes encode a protein complex that is homologous to the bacterial subunits β or β′. The promoter specificity subunit displays no detectable sequence similarity to bacterial sigma factors. **b** The consensus of AR9 late promoters recognized by the nvRNAP. Both DNA strands are shown. **c** The dependence of the AR9 nvRNAP in vitro transcription activity on the position and number of T bases in the promoter, which is located in the template strand of DNA (bold-underlined). **d** The resistance of recombinantly expressed gp226 to proteolysis by trypsin. The identities and sizes of labeled major products, given as residue ranges, have been established using mass spectrometry. FL stands for the full-length protein. Two technical replicates of two biological replicates of the in vitro transcription and trypsin proteolysis experiments resulted in similar outcomes and one of them is shown. The uncropped autoradiograph and SDS PAGE are presented in Supplementary Fig. 1 and Supplementary Data 1.

a result, the AR9 nvRNAP claw is smaller and has a boxier appearance than that of its cellular counterparts. In bacterial enzymes, the α subunit dimer serves as a platform for the assembly of the β and β′ subunits[12]. In the AR9 nvRNAP structure, the split site of the β′ subunit is spatially close to the putative β′-α$^{II}$ interface (Fig. 3), which suggests that this location likely represents a critical point for the formation of tertiary and quaternary structure.

Inside the catalytic cleft, the AR9 nvRNAP core contains all of the structural elements required for catalysis, stabilization of the open promoter complex, and promoter clearance found in multisubunit RNAPs[12,14], except for the β′ rudder (Figs. 2c, d, 3). The related phiKZ nvRNAP is similar to the AR9 nvRNAP in all these aspects, including the absence of the rudder, which thus could represent a characteristic feature of jumbo phage RNAPs (Supplementary Fig. 3). The β′ rudder is a twisted β-hairpin that

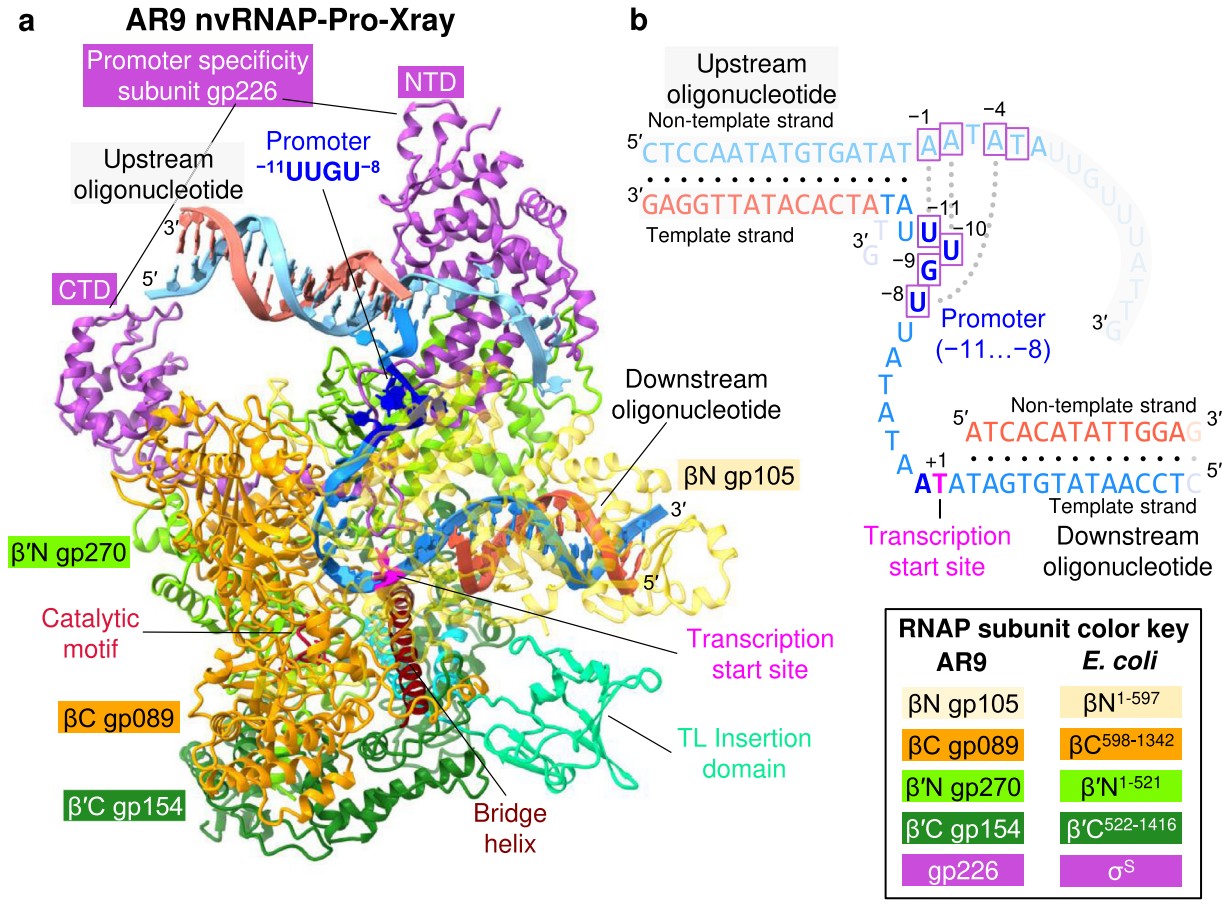

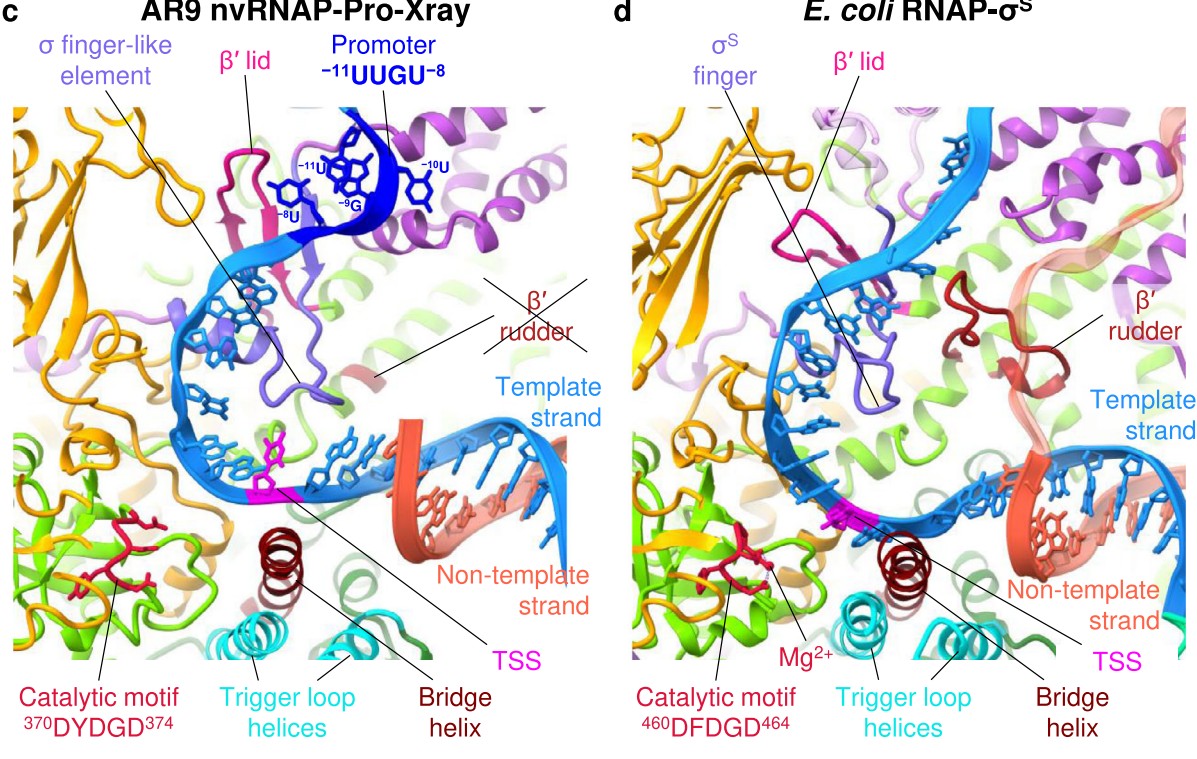

**Fig. 2 Structure of the AR9 nvRNAP promoter complex. a** Ribbon diagram of the crystal structure of the AR9 nvRNAP in complex with a forked oligonucleotide containing the AR9 late promoter P077 in its 3′-overhang region (AR9 nvRNAP-Pro-Xray). Structural elements that are either unique to the AR9 nvRNAP or common to all RNAPs are labeled and color coded. The βN gp105 subunit is semitransparent for clarity. **b** Schematic of the two oligonucleotides that bound to one AR9 nvRNAP molecule resulting in a transcription bubble-like structure. Bases disordered in the crystal structure are rendered semitransparent. Bases in purple boxes interact with the protein. The dashed lines indicate that the sequence of the non-template strand which is in-register with the promoter is partially complementary to it. **c, d** Structure of the catalytic centers of the AR9 nvRNAP and *E. coli* RNAP-σ^S (PDB code 5IPM[21]). Here and elsewhere, TSS stands for the transcription start site. The 2.4 region of σ^S, which is not present in gp226, is rendered semi-transparent. The *E. coli* RNAP-σ^S structure contains a short RNA product that is not shown for clarity. A part of the DNA non-template strand in the *E. coli* RNAP-σ^S structure is semitransparent for clarity.

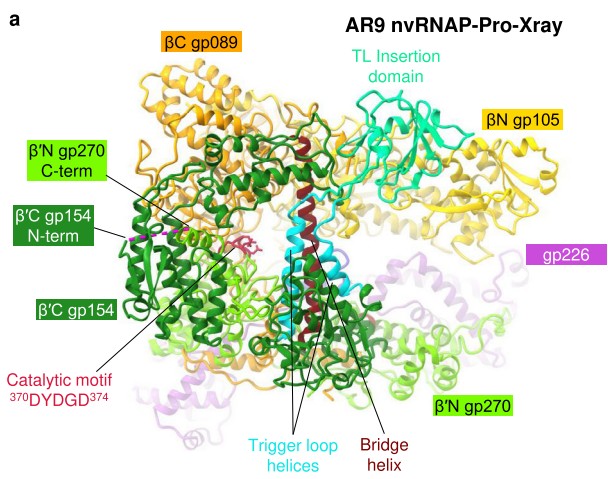

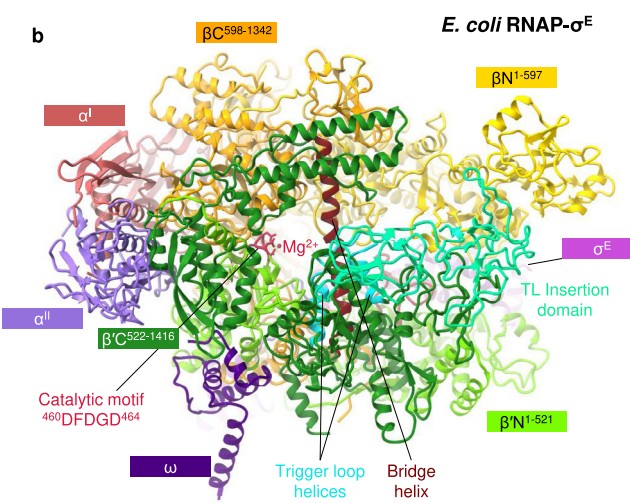

**Fig. 3 Comparison of the AR9 nvRNAP and *E. coli* RNAP-σ^E holoenzymes. a, b** Ribbon diagrams of the AR9 nvRNAP and RNAP-σ^E (PDB code 6JBQ[22]), respectively, are viewed from the NTP entrance channel. Nucleic acids are not shown for clarity. Gp226 and σ^E extend into the plane of the paper and are almost completely obscured by the depth-cueing effect. In both molecules, key elements are colored similarly and labeled. In panel **a**, the color code is as in Fig. 2a.

is present in all known RNAPs. It extends from one of the β′ clamp α-helices and interacts with the RNA-DNA hybrid near the active site[12]. In bacterial RNAPs, deletion of the β′ rudder impairs promoter opening and destabilizes the elongation complex but

does not affect the efficiency of transcription termination or the length of the RNA-DNA hybrid[15]. The elongation complex of the AR9 nvRNAP must be stabilized by a different mechanism.

The conformation of the ^370DYDGD^374 catalytic motif of the AR9 nvRNAP, which is located near the C terminus of the β′N subunit gp270, is similar to that found in other RNAPs. The side chains of the three conserved aspartates are poised to bind a Mg^2+ ion that is universally conserved in all nucleotidyltransferases[16], albeit the resolution of X-ray and cryo-EM data is insufficient for resolving it (Figs. 2c, d, 3a). The electron densities of the template DNA strand at the TSS and that of the complementary region of the non-template strand are also poor. Therefore, the structural basis for the conservation of the TSS-centered part of the promoter consensus element (Fig. 1b) remains to be determined.

Similar to the *E. coli* RNAP[17], the trigger loop (TL) of the AR9 nvRNAP contains an insertion domain (residues 400–508 of β′C gp154, Figs. 2a, 3). The latter element will be further referred to as TLID. The TL undergoes major conformational changes during the catalytic nucleotide addition cycle and template translocation[12,14], and the presence of an TLID in the *E. coli* system has not been fully reconciled with these transformations[18]. The fold of the AR9 nvRNAP TLID is different from that of the *E. coli* RNAP and, in fact, from any protein in the PDB. Its sequence is also unique and found only in nvRNAPs of other jumbo phages[9,10]. Its position in the structure of the nvRNAP core is also different from that of the *E. coli* RNAP.

In the AR9 nvRNAP-core-Xray structure the TLID has located roughly in-between the β and β′ pincers where it partially obstructs the downstream DNA channel (Fig. 4a). The TLID carries a negative charge on its DNA-facing surface (Fig. 4b). Similar to the negatively charged σ1.1 domain of bacterial σ^70 factors[19], the AR9 nvRNAP TLID may function to inhibit non-specific interactions of the enzyme with nucleotide templates.

Out of 10 independent copies of the AR9 nvRNAP core belonging to two different crystal forms of AR9 nvRNAP-core-Xray datasets (Standard and Large unit cell crystals, Supplementary Table 2), the TLID is ordered in only one molecule. In the AR9 nvRNAP-holo-cryoEM structure, the TLID is fully disordered (Fig. 5a, b). Considering the intrinsic propensity of this domain to large motions, it may participate in translocation by sliding on the DNA and exerting a force on the TL.

A comparison of the AR9 nvRNAP-holo-cryoEM and AR9 nvRNAP-Pro-Xray structures shows that the pincers of the AR9 nvRNAP claw are more open in the DNA-free holoenzyme state, although the two conformations are similar. The two structures can be superimposed with an RMSD of 1.77 Å for 2089 out of 2247 (or 93%) Cα atoms participating in the alignment, and the AR9 nvRNAP-Pro-Xray structure fits into the AR9 nvRNAP-holo-cryoEM map as a rigid body with a correlation coefficient of 0.76 (Fig. 5a). The angle between the clamp and lobe in the AR9 nvRNAP-holo-cryoEM structure is about 3° greater than in the AR9 nvRNAP-Pro-Xray structure. This conformational change is in line with the reported closing of the bacterial RNAP claw during promoter binding[20], albeit with a reduced extent.

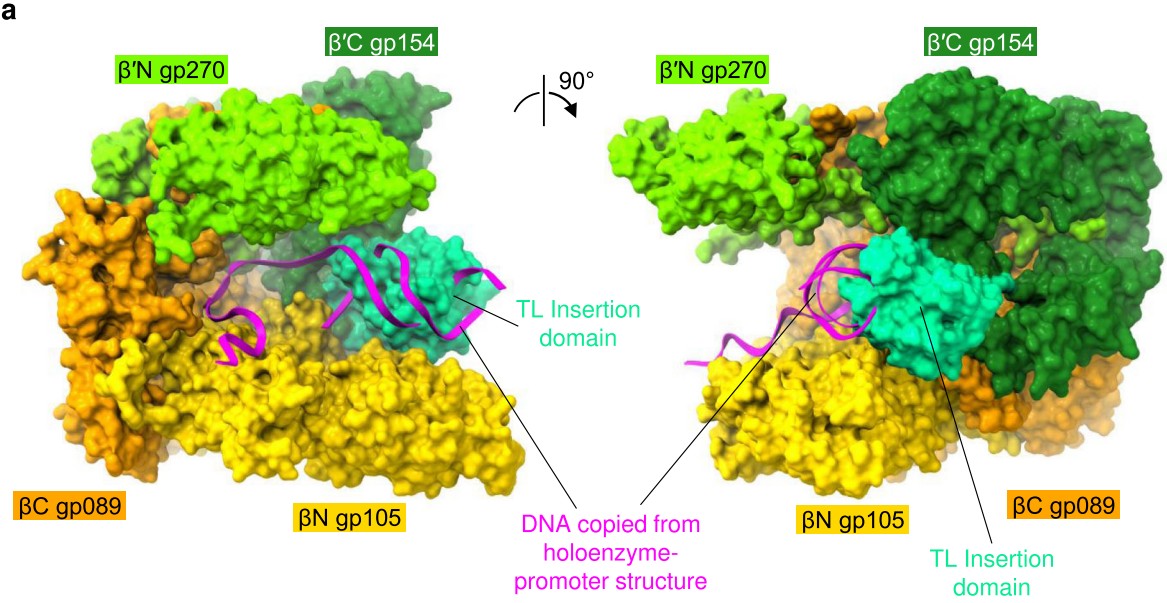

**AR9 nvRNAP-core-Xray**

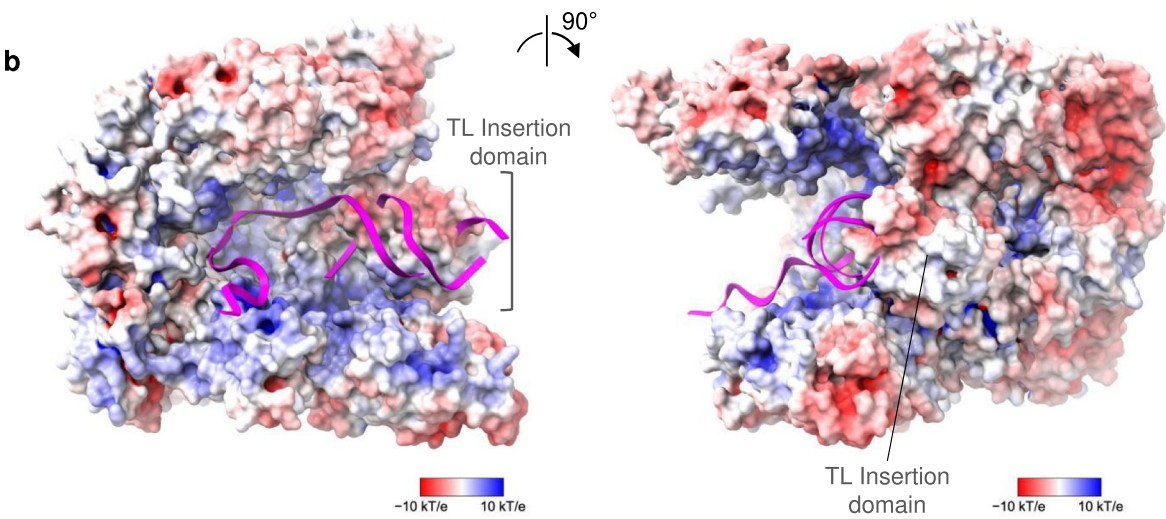

**Fig. 4 Structure of the AR9 nvRNAP core. a** Molecular surface of the AR9 nvRNAP core crystal structure (AR9 nvRNAP-core-Xray), with subunits colored as in Fig. 2a, with the downstream DNA oligonucleotide copied from the holoenzyme-promoter (AR9 nvRNAP-Pro-Xray) structure. The trigger loop insertion domain (TLID) partially obstructs the DNA binding cleft. **b** Electrostatic potential is mapped onto the molecular surface of the AR9 nvRNAP core. Both orientations are as in panel **a**.

**The structure of promoter-specificity subunit gp226**. The AR9 nvRNAP promoter-specificity subunit gp226 consists of two globular domains—a larger N-terminal domain (NTD, residues 1–264) and a smaller C-terminal domain (CTD, residues 295-464)—connected by a linker (Figs. 2a, 6a).

Gp226 interacts with the nvRNAP core in a manner resembling that of bacterial σ factors[11,14,21,22], and all elements that come in contact with the body of the core have structural counterparts in bacterial σ factors. Residues 184-264 of gp226 fold into a σ$_2$-like domain (Fig. 6a), which interacts in a sequence-specific manner with the non-template strand of the −10 promoter element in bacterial σ factors and comprises their most conserved part[23–25] (Fig. 6b). Gp226 residues 265-294 form a σ finger-like structure that invades the catalytic cleft and forms an augmented β-sheet with the β′ lid (Figs. 2c, 6a). Gp226

residues 295-316 comprise an α-helix that matches the N-terminal α-helix of the σ$_4$ domain (Fig. 6a, b).

The most similar σ factor with a known structure, the *E. coli* σ$^E$, displays a Cα-Cα RMSD of 4.2 Å and a sequence identity of 9.6% when superimposed onto residues 184–316 of gp226 which comprise its σ$_2$-, finger- and σ$_4$-like elements. Although the structural and sequence similarities are low, the σ-like part of gp226 spans a contiguous region of 133 residues and is nearly equal in size to the entire σ$^E$ structure (Fig. 6a, b). Furthermore, gp226 and bacterial σ factors interact with the template DNA in a similar manner resulting in similar transcription bubbles. Therefore, despite the vanishingly low sequence conservation, gp226 is a likely homolog of bacterial σ factors. The peripheral parts of the gp226 NTD and CTD have been replaced with new folds but all elements that interact with the body of the enzyme and with

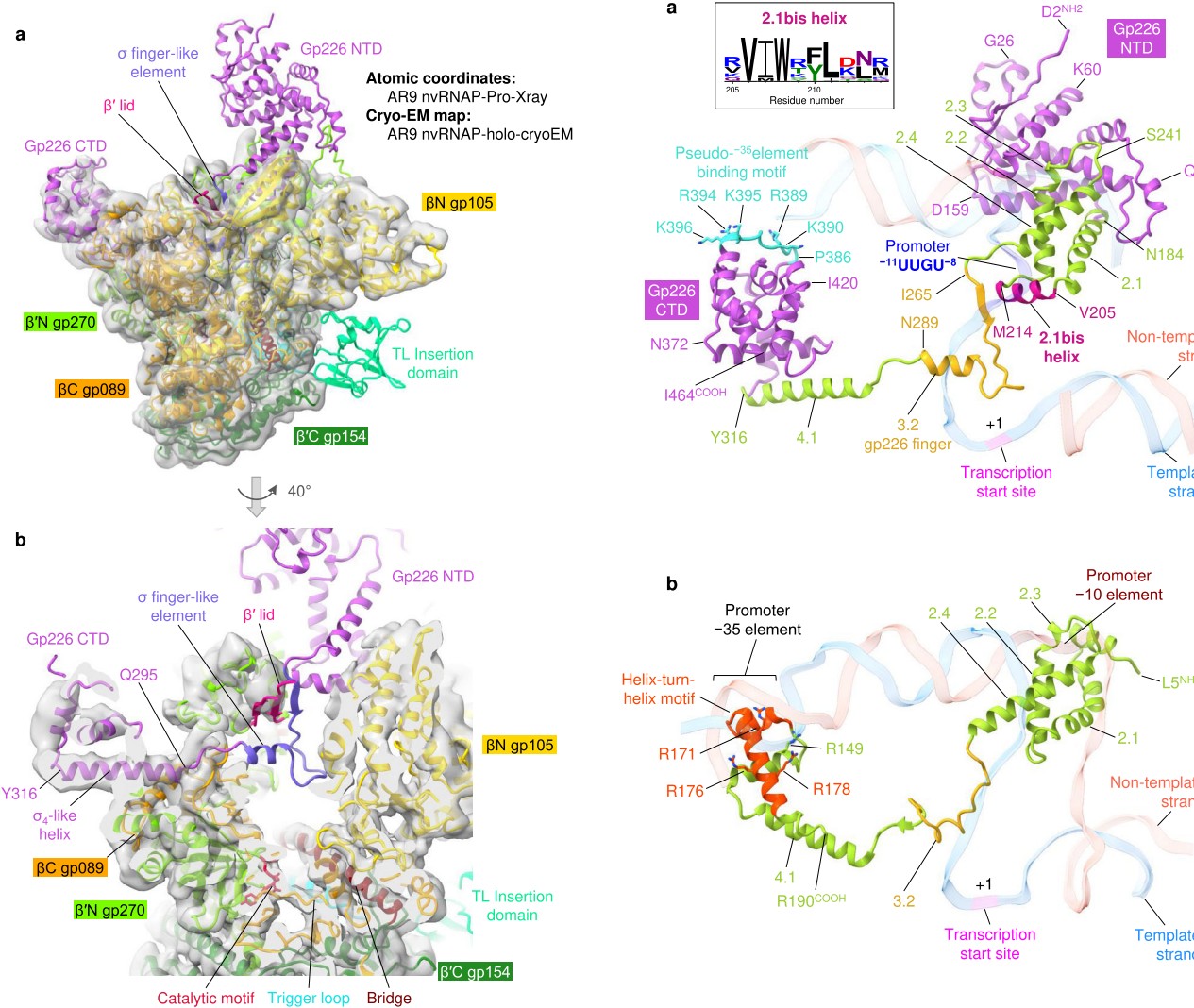

**Fig. 5 Cryo-EM structure of the AR9 nvRNAP holoenzyme. a** Cryo-EM map of the AR9 nvRNAP holoenzyme (AR9 nvRNAP-holo-cryoEM) contoured at 4.0 std dev above the mean (semitransparent gray) with the fitted as a rigid body atomic model of the AR9 nvRNAP promoter complex sans DNA (AR9 nvRNAP-Pro-Xray) and is colored as in Fig. 2a. **b** A zoomed-in view of the catalytic cleft demonstrating the degree of gp226 disorder.

**Fig. 6 Structure of the promoter specificity subunit gp226. a** Ribbon diagram of gp226 with regions structurally similar to bacterial σ factors colored in yellow green (helices 2.1 through 2.4 and 4.1) and gold (finger). The unique N- and C-terminal domains colored medium orchid. Residue numbers and identities are given at key locations. The DNA strands are colored as in Fig. 2a, b and are semitransparent. The pseudo−35-element binding motif is turquoise and its positively charged and solvent-exposed residues are shown in a stick representation. Inset: Conservation of residues comprising the 2.1bis helix. WebLogo[78] shows a 2.1bis helix-centered fragment of a full-length multiple sequence alignment, calculated using ClustalX[77], of twelve gp226 homologs currently available at GenBank. **b** Ribbon diagram of the *E. coli* σ[E] factor (PDB code 6JBQ[22]) with its helix-turn-helix motif colored in orange red. Positively charged residues that interact with the DNA are shown in a stick representation and labeled. The DNA backbone is semitransparent. .

DNA (albeit with small modifications for the latter) have been retained.

A Protein Data Bank-wide search[26] for folds resembling that of the gp226 NTD and CTD resulted in a single definitive match. The CTD of gp68, a subunit of the phage phiKZ nvRNAP[6,8], can be superimposed onto the gp226 CTD with an RMSD of 3.2 Å for 142 equivalent Cα atoms (out of 173) and a sequence identity of 11% (Supplementary Fig. 4a). As the rest of the gp68 structure (residues 1-303) is disordered, we used AlphaFold[27] Colab to model it (the atomic coordinates are given in Supplementary Data 2). The local distance difference test[28] of this model for residues 1-277 was 85.3, indicating a very high level of confidence (Supplementary Fig. 4b). This model can be superimposed onto the gp226 NTD with an RMSD of 3.0 Å for 198 equivalent Cα atoms (out of 277) and a sequence identity of 8.6% (Supplementary Fig. 4a). Thus, even though gp226 is as divergent from gp68 sequence-wise as it is from bacterial σ factors, their similarities in

structure and location within the RNAP holoenzyme complex suggest that all these proteins have a common ancestor. Functionally, gp68 of phiKZ appears to be more closely related to bacterial σ factors than gp226 of AR9 because the phiKZ nvRNAP recognizes T-containing dsDNA[6]. On the other hand, phiKZ gp68 is required for the assembly of an elongation-capable phiKZ nvRNAP complex[8], whereas an elongation-capable AR9 nvRNAP core assembles without gp226[3]. The latter property of gp226 is in line with that of bacterial σ factors.

**Structural basis of template strand promoter recognition**. The unique, template strand-specific mode of promoter recognition by AR9 nvRNAP is enabled by a few small adaptations in the $\sigma_2$- and σ finger-like elements of gp226 (Fig. 6). A tight turn connecting helices 2.1 and 2.2 in bacterial σ factors is replaced by a short α-helix in gp226 (residues 205–214). This 2.1bis helix creates a bridge linking the two pincers of the AR9 nvRNAP claw that, together with the βN gp105 subunit, comprises a binding site for the promoter in the template strand of DNA (Fig. 6a). The gp226 finger forms an augmented β sheet with the β' lid (residues 161-177 of β'N gp270) such that the β' lid reaches the −8 position uracil base of the $3'-^{-11}UUGU^{-8}-5'$ promoter motif and tucks it in against the 2.1bix helix (Fig. 2c, d). The β' lid of AR9 nvRNAP is three residues longer than its bacterial counterpart, which further enhances and facilitates this interaction (Fig. 2c).

**Promoter DNA structure and the design of a T-specific enzyme**. The structure of the −10-like $3'-^{-11}UUGU^{-8}-5'$ template strand promoter motif is fully resolved in the electron density map of the AR9 nvRNAP-Pro-Xray dataset (Fig. 7a). The $^{-8}U$ nucleotide is partially disordered in the AR9 nvRNAP-Pro-cryoEM map (Fig. 7b). The most critical and obligatory $^{-10}U$ base, the replacement of which by a T abolishes promoter recognition[3] (Fig. 1c), is buried in a deep pocket at the interface of the gp226 2.1bis helix and βN gp105 (Figs. 2a, c, 8a). In this pocket, the $^{-10}U$ base is wedged between the side chains of gp226 I207 and βN gp105 R363 (Fig. 8a), forming a stacking interaction with the latter. Its Watson-Crick interface forms hydrogen bonds with the side chain of βN gp105 K375 and with the main chain N of gp226 V206 and I207. Most importantly, the C5 atom of the $^{-10}U$ pyrimidine ring is only 3.9 Å away from the Cβ of V206, suggesting that a C5 position methyl group would clash with the V206 side chain (Fig. 8a). Accordingly, a holoenzyme containing gp226 with a V206G substitution recognized $^{-10}U$- and $^{-10}T$-containing promoters with equal efficiencies (Fig. 8b, Supplementary Fig. 1, Supplementary Data 1). Notably, all close homologs of gp226 proteins in jumbo phages with deoxyuridine-containing genomic DNA[9,10] display high sequence conservation of the 2.1bis helix with the critical valine being absolutely conserved (Fig. 6a inset, Supplementary Fig. 5). This suggests that these phages employ a common mechanism for uracil-dependent promoter recognition.

The requirement of U vs. T in the −11th position of the promoter is nearly as strong as in the −10th position. In addition, a G is required in the −9th position[3]. However, the enzyme displays almost no U vs. T preference in the −8th position (Fig. 1c). In the promoter complex, the bases of $^{-11}U$ and $^{-9}G$ form a stacking interaction such that the C5 and C6 atoms of $^{-11}U$ butt against the sugar-phosphate backbone of the $^{-11}UUG^{-9}$ segment, leaving no space for a C5 position methyl group (Fig. 8a). There is a stacking interaction between $^{-9}G$ and the phenol ring of gp226 Y210 (which belongs to the 2.1bis helix), and there are three hydrogen bonds between the Watson-Crick interface of $^{-9}G$ and the main chain of gp226 residues F261 and Y263, which provides a rationale for the G requirement in this position. $^{-8}U$ forms one hydrogen bond with $^{-11}U$ and one hydrogen bond with the tip of the β' lid, which is longer than in its bacterial counterparts, as described above. The C5 position of $^{-8}U$ points into the solution and can accommodate the additional methyl group of T (Fig. 8a).

**Promoter-complementary DNA motif interacts with gp226 NTD**. The gp226 NTD displays several deep pockets that capture the ss part of the upstream oligonucleotide, which mimics the

non-template strand of the transcription bubble (Figs. 7c, 8c). In the transcription bubble (Fig. 2b), this part of the non-template strand must have a sequence complementary to the template strand promoter motif ($5'-^{-11}AACA^{-8}-3'$, the numbering is relative to the TSS). Our oligonucleotide contained a similar motif in its ss part ($5'-^{-1}AATA^{-4}-3'$, same numbering as the downstream nucleotide, Fig. 2b). Together with its neighboring bases, this sequence matched the appearance of the electron density (Fig. 7c). The base of the mismatched third position nucleotide (T↔C) does not interact with the gp226 NTD but instead protrudes into the solution (Figs. 7c, 8c). Despite being only partially complementary to the promoter, this motif was likely a key determinant in the fortuitous binding of the upstream oligonucleotide.

The gp226 NTD interacts with the backbone and bases of the non-template DNA strand of the transcription bubble via π-π stacking, ion pairs, and hydrogen bonds (Fig. 8c). The length of this interface exceeds 30 Å. The extent of these interactions suggests that they play an important role in promoter recognition, the unwinding of the dsDNA template, and the stabilization of the transcription bubble. Indeed, a Y246A substitution, which eliminated pi-pi stacking between the side chain of Y246 and the $^{-2}A$ base of the $^{-1}AATA^{-4}$ motif, abolished transcription on dsDNA but did not affect transcription on a fork template (Fig. 8c, d, Supplementary Fig. 1, Supplementary Data 1). Furthermore, a S245E substitution introduced a large, negatively charged side chain on the surface of the gp226 NTD that interfered with the trace and conformation of the sugar-phosphate backbone between $^{-2}A$ and $^{-4}A$ (Fig. 8c). As a consequence, the transcriptional activity of the holoenzyme containing the S245E gp226 mutant on a dsDNA template was weak, whereas its fork template activity was at or above that of WT (Fig. 8d, Supplementary Fig. 1, Supplementary Data 1).

In the AR9 nvRNAP-holo-cryoEM structure, the NTD and σ finger-like element of gp226 are disordered (Fig. 5a, b). Similar to the TLID, the disorder is likely due to positional heterogeneity since both the gp226 NTD and TLID possess well-defined hydrophobic cores and are folded in other states of the nvRNAP complex. Furthermore, the NTD is resistant to proteolysis by trypsin in gp226 recombinantly expressed on its own (Fig. 1d). The order-disorder transitions of the gp226 NTD and σ finger-like element play a role in the promoter recognition mechanism described below.

**Interaction of gp226 CTD with dsDNA is sequence independent**. Although weak, the X-ray electron density and cryo-EM map of the upstream oligonucleotide stretches from the $\sigma_2$-like part of the gp226 NTD to the gp226 CTD (Fig. 7c, d). This interaction is about 35 DNA base pairs upstream from the TSS drawing a parallel to the −35 consensus element of bipartite bacterial promoters. The recognition of the −35 element by bacterial σ factors is mediated by a helix-turn-helix motif of the $\sigma_4$ domain[29] (Fig. 6b), which interacts with the major groove of dsDNA[30] in a sequence-specific manner. AR9 nvRNAP promoters, however, display no sequence conservation in this region (Fig. 1b) and, accordingly, the gp226 CTD interacts with the minor groove of dsDNA which displays few sequence-specific features in the B form[31]. Furthermore, this interaction is mediated by a scrunched β-strand (Fig. 6a) that carries several positively charged residues (R389, K390, R394, K395, K396), but not by a helix-turn-helix motif, which is absent from the gp226 structure. We termed the DNA interacting element of the gp226 CTD (amino acids 386-395) a pseudo-$^{-35}$element-binding motif.

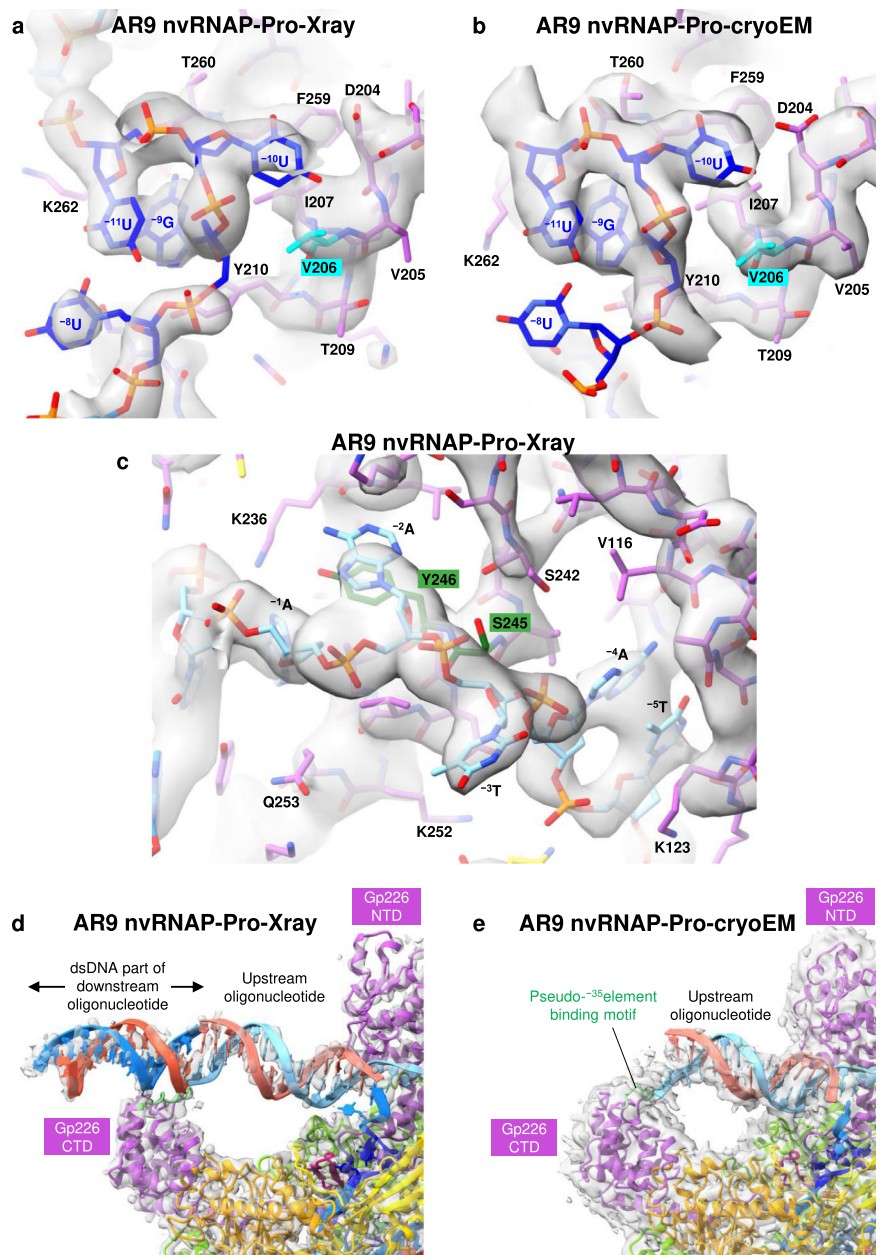

**Fig. 7 Electron density of the AR9 nvRNAP-DNA interacting regions. a**, **b** Composite omit X-ray electron density and cryo-EM map of the promoter binding pocket with refined atomic models. The X-ray and cryo-EM maps are contoured at 1.5 and 5.0 std dev above the mean, respectively. The carbon atoms are colored as in Fig. 2a. V206, which is critical for U specificity, is colored cyan. **c** A fragment of composite omit X-ray electron density that is interpreted in terms of non-template strand nucleotides that are (fortuitously) partially complementary to the template strand promoter sequence. The X-ray map is contoured at 1.5 std dev above the mean. The carbon atoms are colored as in Fig. 2a. Residues Y246 and S245 that affect recognition of dsDNA template are colored forest green. **d**, **e** Composite omit X-ray electron density and cryo-EM map of gp226 CTD in the AR9 nvRNAP promoter complex. The cryo-EM and X-ray maps are contoured at 1.0 and 2.0 std dev above the mean, respectively. The ds segment of the downstream oligonucleotide belonging to a neighboring unit cell is shown in the X-ray map (see Supplementary Fig. 2c). The cryo-EM map of the upstream oligonucleotide is of insufficient quality for model building and its atomic model has been copied from the AR9 nvRNAP-Pro-Xray structure for illustration purposes only. Proteins are colored as in Fig. 2a. The pseudo-$-35$element binding motif is colored light green.

To examine the role of the pseudo-$-35$element-binding motif in promoter recognition, we removed most of the positive charge displayed on its surface by replacing R389, K390, R394, K395, and K396 of gp226 with alanines (we called this mutant A[5]) (Fig. 8e, f). As the AR9 nvRNAP holoenzyme containing A[5] gp226 had a lower activity overall, we compared its activity to that of the wild type (WT) holoenzyme on two dsDNA templates that either contained or lacked the upstream part required for interaction with the pseudo-$-35$element-binding motif (the $[-60, +80]$ and

$[-16, +80]$ templates in Fig. 8g, respectively, see also Supplementary Fig. 1, Supplementary Data 1). The WT enzyme exhibited a greater decrease in activity on the $[-16, +80]$ template compared to that of the mutant, which shows that the pseudo-$-35$element-binding motif is essential for optimal promoter recognition.

The AR9 gp226 pseudo-$-35$element-binding motif maps onto a disordered part of phage phiKZ gp68[8] (residues 413–429, Supplementary Fig. 3a). This region of gp68's surface carries a

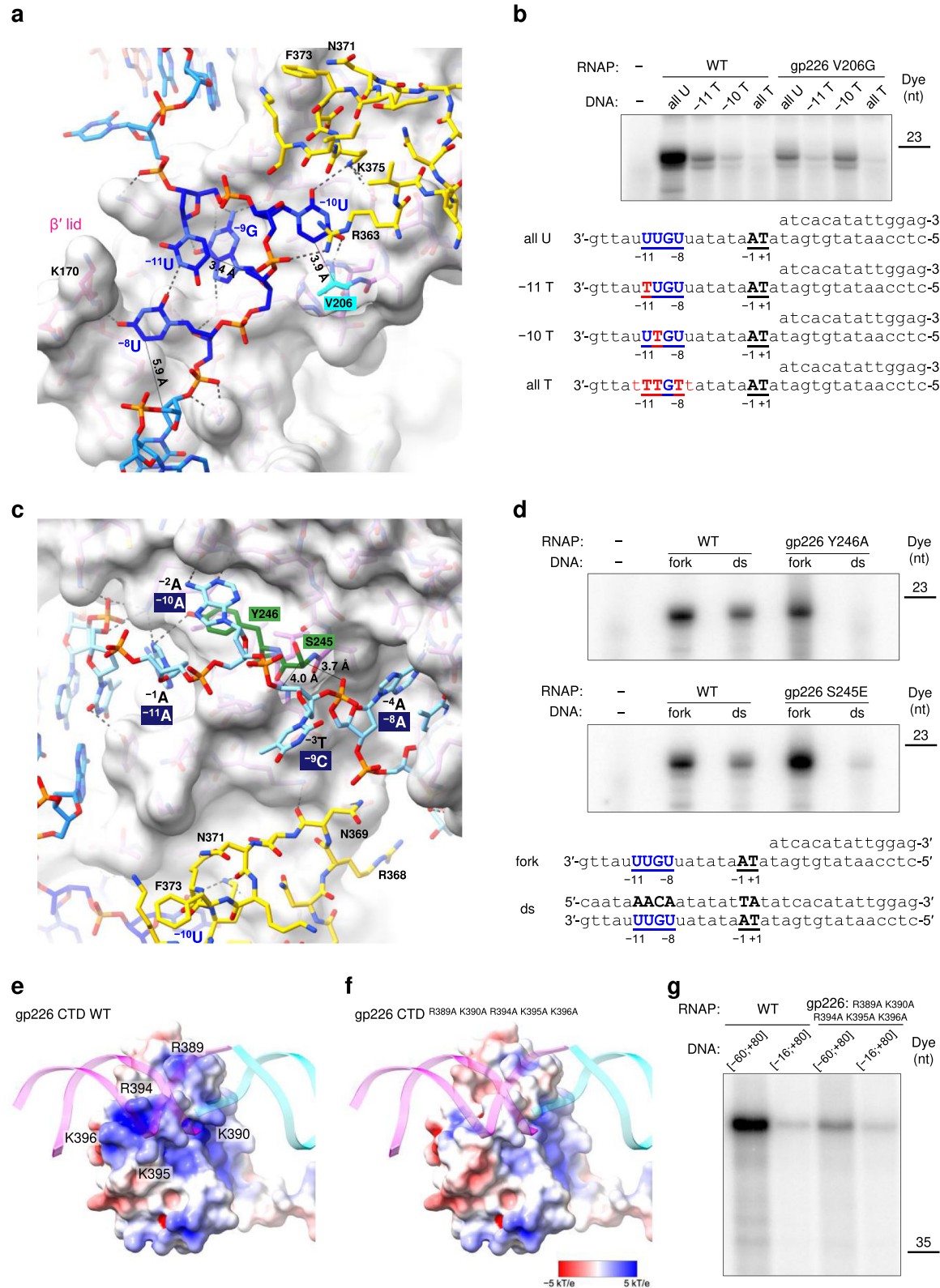

positive charge, akin to that of AR9 gp226 (Fig. 8e), even without the inclusion of disordered residues in electrostatic potential calculations. Again, analogous to the AR9 nvRNAP, phiKZ nvRNAP promoters show no sequence conservation 35 nucleotides upstream of the TSS[5]. Considering (i) the homology of the

phiKZ and AR9 nvRNAPs to cellular RNAPs[8], (ii) the homology of AR9 gp226 to phiKZ gp68 described above, and iii) the presence of positive charges at equivalent locations on the surface of the AR9 gp226 and phiKZ gp68 CTDs, the phiKZ nvRNAP is thus likely to form an AR9 nvRNAP-like transcription bubble in

**Fig. 8 Interaction of the AR9 nvRNAP with DNA in the AR9 nvRNAP-Pro-Xray promoter complex. a** Atomic model of the AR9 nvRNAP promoter recognition element. Gp226 is shown as a semitransparent molecular surface. Only a small fragment of βN gp105 that participates in the formation of the $^{-10}$U binding pocket is shown for clarity. Interchain and DNA-intrachain hydrogen bonds are shown as dashed lines. Thin, straight lines connect the C5 atom of the uracil pyrimidine ring to the closest protein or DNA atoms that lie in-plane with the ring. The carbon atoms are colored as in Fig. 2a, except for V206, which is shown in cyan. **b** The in vitro transcription activity of the AR9 nvRNAP holoenzyme containing wild type (WT) gp226 or the V206G gp226 mutant has been tested using various U- and T-containing templates. **c** Interaction of the upstream oligonucleotide with the gp226 NTD. The same gp226 and βN gp105 fragments as in Fig. 8a are shown but are tilted to improve visibility. The nucleotides are numbered according to the original nomenclature of the template strand (Fig. 2b). The putative base identities and their numbers relative to the TSS (as would be found in a dsDNA transcription bubble) are given in midnight blue colored boxes. **d** The in vitro transcription activity of the AR9 nvRNAP holoenzyme containing gp226 mutants with an altered structure of the non-template strand binding groove. **e, f** The surface electrostatic potentials of the pseudo$^{-35}$-element binding motif in WT gp226 and the A$^5$ gp226 mutant. **g** The ssDNA and dsDNA in vitro transcription activities of the AR9 nvRNAP holoenzyme containing WT gp226 and the A$^5$ gp226 mutant. For each in vitro transcription experiment, two technical replicates of two biological replicates resulted in similar outcomes and one of them is shown. The uncropped autoradiographs are presented in Supplementary Fig. 1 and Supplementary Data 1.

which the CTD of gp68 participates in the binding of upstream dsDNA. Furthermore, as all these properties are seemingly conserved for such distantly related viruses as AR9 and phiKZ that infect unrelated hosts (Gram-positive *B. subtilis* and Gram-negative *Pseudomonas aeruginosa*, respectively), and have different genomic DNA base composition[1,32] and genome replication strategies[33], the supposition of AR9 nvRNAP-like transcription bubbles can be extended to nvRNAPs of all jumbo phages.

**Free energy of template strand promoter binding.** To reconcile the tight integration of AR9 nvRNAP promoter DNA into the promoter complex (Figs. 2a, c, 8a) with the transient nature of this complex, we examined the binding free energy of the three best-ordered promoter nucleotides 3′-$^{-11}$UUG$^{-9}$-5′ to the AR9 nvRNAP holoenzyme by executing a double decoupling method molecular dynamics protocol[34–36] (Supplementary Data 3). The procedure assumes that the conformation of the enzyme does not appreciably change upon promoter binding. As such, the simulations describe a state in which the gp226 NTD has associated with the AR9 nvRNAP core.

The standard binding free energy was calculated by combining the results of four separate simulations that corresponded to the vertical reactions in the thermodynamic cycle shown in Fig. 9a. After the equilibration of the system (Fig. 9b, c), two types of simulations were performed: (i) "alchemical transformations" in which the occupancy of the oligonucleotide located either in the promoter pocket ($\Delta G_{\text{alchemical}}^{\text{bound}}$) or in bulk water ($\Delta G_{\text{alchemical}}^{\text{bulk water}}$) was reduced to zero while the oligonucleotide was harmonically constrained to maintain the promoter pocket bound conformation (Fig. 9d, e), and (ii) calculations of the entropic cost of such harmonic constraints for an oligonucleotide located in the promoter pocket ($\Delta G_{\text{restrain}}^{\text{bound}}$) (Fig. 9f–l) and in bulk water ($\Delta G_{\text{restrain}}^{\text{bulk water}}$) (Fig. 9m). To ensure reproducibility and to minimize bias, all simulations were run bidirectionally.

The favorable energetics of promoter binding via alchemical transformation ($-12.7 \pm 2.3$ kcal/mol, Fig. 9d, e) are partially offset by the unfavorable entropic contributions of constraints on DNA conformation and position ($5.8 \pm 1.5$ kcal/mol, Fig. 9f–m). The resulting free energy gain upon complex formation is $-6.9 \pm 2.8$ kcal/mol, which shows that the interaction of this promoter element with the enzyme is relatively weak. Thus, despite its unusual structure in which the $^{-10}$U is buried in a deep pocket and $^{-9}$G and $^{-11}$U form a stacking interaction, the promoter complex is transient, and the enzyme can easily proceed towards elongation.

**Mechanism of template strand promoter recognition in dsDNA.** Combining these findings, we propose the following model for promoter recognition by the AR9 nvRNAP (Fig. 10). In the free state of the AR9 nvRNAP holoenzyme molecule, the NTD of gp226 is folded but does not interact with the body of the enzyme (it is positionally disordered or mobile) and the promoter-binding pocket is absent (Fig. 5). The NTD of gp226 is attached to the CTD and the core via a linker that will eventually form a σ finger-like structure in the promoter complex. The enzyme displays two positively charged surface patches that have DNA binding propensity—the pseudo-$^{-35}$element-binding motif and a patch on the gp226 NTD surface that interacts with the promoter-complementary motif in the non-template DNA strand (Supplementary Fig. 6 and State 1 in Fig. 10).

The process of promoter recognition contains the following steps. (1) The gp226 NTD captures the non-template strand in a groove on its surface by recognizing a partially promoter-complementary motif and enabling the initial melting of template dsDNA (State 2 in Fig. 10). Both events are facilitated by the AU-rich promoter-containing regions, which may transiently display flipped-out bases. Only three bases of this motif interact with the protein (the third position base $^{-9}$C does not, Fig. 8c). The prevalence of the 5′-AANA-3′ motif in the AR9 genome (18,972 instances or every ~26 nucleotides) may allow the enzyme to scan the template. (2) The pseudo-$^{-35}$element-binding motif of the gp226 CTD interacts with dsDNA, reducing the conformational space available to the gp226 NTD, and thus promoting its binding to the body of the enzyme (State 3 in Fig. 10). (3) The NTD of gp226 comes in contact with the body of the enzyme, fully separating the DNA strands, forming a σ finger-like element and a transcription bubble, and placing the template strand at the [gp226]:[βN gp105] interface. This interface captures a flipped-out $^{-10}$U base and buries it into the $^{-10}$U recognition pocket. Simultaneously, the DNA strand is squeezed slightly such that the bases flanking the flipped-out $^{-10}$U base form a stack, and the identities of the stacked $^{-9}$G and $^{-11}$U bases are verified via geometry-sensitive interactions (hydrogen bonds and ion pairs). Additional interactions are formed at the catalytic center where the TSS is recognized (State 4 in Fig. 10). As the free energy of promoter recognition is nevertheless reasonably low and the conformation of the sugar-phosphate backbone for the four nucleotides of the promoter motif is close to that of dsDNA, the enzyme can efficiently proceed with elongation.

Here, we have explained the functional mechanism of a phage-encoded RNAP that contains a unique promoter-specificity subunit, recognizes the promoter in the template strand of DNA, requires uracil bases in the promoter, and does not use a common helix-turn-helix motif for the binding of dsDNA. Even

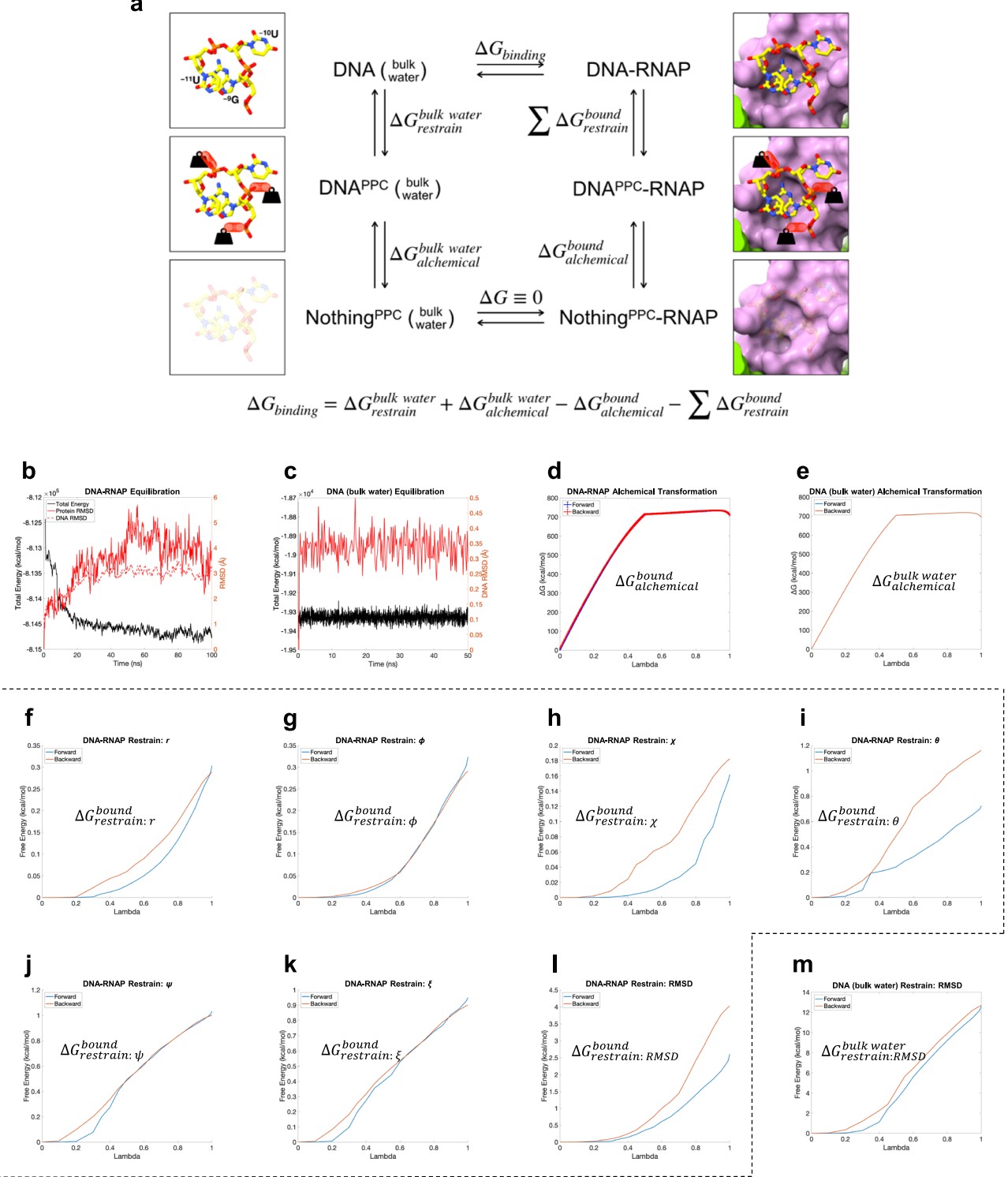

**Fig. 9 Derivation of the promoter binding free energy using molecular dynamics. a** Thermodynamic cycle of promoter binding. The PPC superscript (e.g. $DNA^{PPC}$) stands for Promoter Pocket Conformation in regard to the structure of the $3'$-$^{-11}UUG^{-9}$-$5'$ DNA trinucleotide. **b** Equilibration and relaxation of the cryo-EM derived atomic model of the AR9 nvRNAP holoenzyme with the $3'$-$^{-11}UUG^{-9}$-$5'$ DNA trinucleotide bound to the promoter pocket.
**c** Equilibration and relaxation of the $3'$-$^{-11}UUG^{-9}$-$5'$ trinucleotide in bulk water. **d**, **e** Energetics of forward and backward alchemical transformations of the $3'$-$^{-11}UUG^{-9}$-$5'$ DNA trinucleotide in the promoter pocket of the AR9 nvRNAP holoenzyme and in the PPC in bulk water, respectively. **f–l** Entropic cost of applying seven harmonic constraints to the $3'$-$^{-11}UUG^{-9}$-$5'$ DNA trinucleotide to maintain it in the promoter pocket-bound state. **m** Entropic cost of the harmonic RMSD constraint on the $3'$-$^{-11}UUG^{-9}$-$5'$ DNA trinucleotide to maintain the PPC.

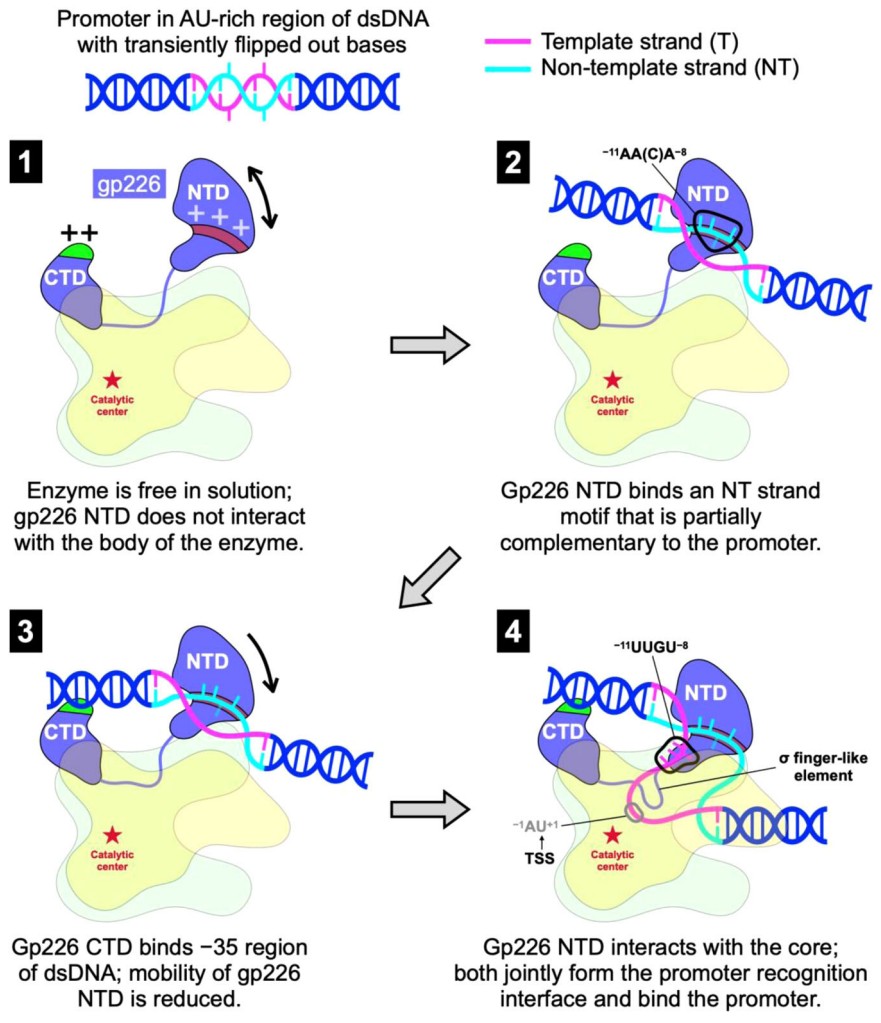

**Fig. 10 The mechanism of template strand promoter recognition in dsDNA.** For clarity, both proteins comprising the N- and C-terminal parts of the β and β' subunits are shown in the same color (light yellow for β and light green for β'). Nucleotide bases are displayed as short sticks. The putatively degenerate, third position base $^{-9}$C of the promoter-complementary motif $^{-11}$AA(C)A$^{-8}$ is shown as a semitransparent stick. The '+' signs indicate positively charged elements of the gp226 NTD and CTD molecular surface that participate in DNA binding. See the main text for full explanation.

though the AR9 nvRNAP and its promoter specificity subunit appear to have a common ancestor with their bacterial counterparts, the extent to which the AR9 nvRNAP promoter recognition mechanism is different from that of any known RNAP shows that our knowledge of the structure and function of these complex enzymes is far from complete.

## Methods

No statistical methods were used to predetermine sample size. The experiments were not randomized and the investigators were not blinded to allocation during experiments and outcome assessment.

**Cloning of the AR9 nvRNAP and its mutants.** Four gene-Blocks (gBlocks) encoding AR9 nvRNAP core enzyme genes optimized for expression in *E. coli* were synthesized by Integrated DNA Technologies (IDT). These gBlocks were assembled into an expression vector on the pETDuet-1 plasmid backbone with the help of the NEBuilder HiFi DNA Assembly Master Mix (New England Biolabs). First, pETDuet-1 was digested by the NcoI and BamHI endonucleases, and gBlocks coding for N-terminally hexahistidine-tagged gp270 and gp154 were ligated. Then, this plasmid was digested by the BglII and XhoI endonucleases and ligated with two gBlocks coding for gp105 and gp089. The resulting plasmid encoded the AR9 nvRNAP core enzyme.

The plasmid for expression of the AR9 nvRNAP holoenzyme was created by inserting an *E. coli*-optimized gp226 gBlock (also synthesized by ITD) into the AR9 nvRNAP core plasmid described above, which was linearized at the XhoI site. This plasmid was used as a template to create mutant versions of the AR9 nvRNAP

holoenzyme by site-directed mutagenesis (the list of corresponding primers is in Supplementary Table 3).

The plasmid encoding the tagless AR9 nvRNAP core enzyme was derived from the His-tagged AR9 nvRNAP core plasmid described above. First, a fragment that contained all the four genes but excluded the Hig-tag was PCR amplified (the primers are listed in Supplementary Table 3). Then, the pETDuet-1 vector was linearized by the NcoI and XhoI endonucleases. A new plasmid was then created by ligating the PCR fragment and the linearized pETDuet-1 vector using the NEBuilder HiFi DNA Assembly Master Mix (New England Biolabs).

In all plasmids, a T7 RNAP promoter, a *lac* operator, and a ribosome binding site were located at appropriate positions upstream of each gene.

**Purification of recombinant AR9 nvRNAP.** Plasmids encoding AR9 nvRNAP core, tagless AR9 nvRNAP core, and holoenzyme or its mutants were transformed into BL21 Star (DE3) chemically competent *E. coli* cells. The cultures (3 L) were grown at 37 °C to OD$_{600}$ of 0.7 in LB medium supplemented with ampicillin at a concentration of 100 μg/mL, and recombinant protein overexpression was induced with 1 mM IPTG for 4 h.

Cells containing over-expressed AR9 nvRNAP holoenzyme or its mutants were harvested by centrifugation and disrupted by sonication in buffer A (40 mM Tris-HCl pH 8, 300 mM NaCl, 3 mM β-mercaptoethanol) followed by centrifugation at 15,000 g for 30 min. Cleared lysate was loaded onto a 5 mL HisTrap sepharose HP column (GE Healthcare) equilibrated with buffer A. The column was washed with buffer A supplemented with 20 mM Imidazole. The protein was eluted with a linear 0–0.5 M Imidazole gradient in buffer A. Fractions containing AR9 nvRNAP holoenzyme or its mutants were combined and diluted with buffer B (40 mM Tris-HCl pH 8, 0.5 mM EDTA, 1 mM DTT, 5% glycerol) to the 50 mM NaCl final concentration and loaded on equilibrated 5 mL HiTrap Heparin-sepharose HP column (GE Healthcare). The protein was eluted with a linear 0–1 M NaCl gradient

in buffer B. Fractions containing AR9 nvRNAP holoenzyme or its mutants were pooled and concentrated (Amicon Ultra-4 Centrifugal Filter Unit with Ultracel-50 membrane, EMD Millipore) to a final concentration of 3 mg/mL, then glycerol was added up to 50% to the sample for storage at −20 °C (the samples were used for transcription assays).

Samples used for crystallization and cryo-EM were produced by following a slightly different procedure. Cells containing over-expressed recombinant AR9 nvRNAP core or holoenzyme were harvested by centrifugation and disrupted by sonication in buffer C (50 mM NaH$_2$PO$_4$ pH 8, 300 mM NaCl, 3 mM β-mercaptoethanol, 0.1 mM PMSF) followed by centrifugation at 15,000 × $g$ for 30 min. Cleared lysate was loaded on 5 mL Ni-NTA column (Qiagen) equilibrated with buffer C, washed with 5 column volumes of buffer C and with 5 column volumes of buffer C containing 20 mM Imidazole. Then, elution with buffer C containing 200 mM Imidazole was carried out. Fractions containing AR9 nvRNAP core or holoenzyme were pooled and diluted ten times by buffer D (20 mM Tris pH 8, 0.5 mM EDTA, 1 mM DTT) or by buffer E (20 mM Bis-tris propane pH 6.8, 0.5 mM EDTA, 1 mM DTT) correspondingly and applied to a MonoQ 10/100 column (GE Healthcare). Bound proteins were eluted with a linear 0.25–0.45 M NaCl gradient in buffer D or E correspondingly.

Cells containing over-expressed recombinant tagless AR9 nvRNAP core were harvested by centrifugation and disrupted by sonication in buffer B followed by centrifugation at 15,000 × $g$ for 30 min. An 8% polyethyleneimine (PEI) solution (pH 8.0) was added with stirring to the cleared lysate to the final concentration of 0.8%. The resulting suspension was incubated on ice for 1 h and centrifuged at 10,000 × $g$ for 15 min. The supernatant was removed and the pellet was resuspended in buffer B containing 0.3 M NaCl. After 10 min incubation, the PEI pellet was formed by centrifugation as previously. Then, the supernatant was removed and the pellet was resuspended in buffer B containing 1 M NaCl followed by centrifugation at 10,000 × $g$ for 15 min. Eluted proteins were precipitated in the supernatant by addition of ammonium sulfate to 67% saturation and dissolved in buffer D and loaded on an equilibrated 5 mL HiTrap Heparin-sepharose HP column (GE Healthcare). The protein was eluted with a linear 0–1 M NaCl gradient in buffer D. Fractions containing tagless AR9 nvRNAP core were pooled and subjected to anion exchange chromatography as described above for AR9 nvRNAP core.

The AR9 nvRNAP core sample was polished and buffer-exchanged using size exclusion chromatography on a Superdex 200 10/300 (GE Healthcare) column equilibrated with buffer D containing 100 mM NaCl. The tagless AR9 nvRNAP core and AR9 nvRNAP holoenzyme were not subjected to size exclusion chromatography—salt concentration in the sample was lowered during the concentration procedure.

The fractions containing AR9 nvRNAP core, tagless AR9 nvRNAP core or holoenzyme were concentrated to a final concentration of 20 mg/mL and used for crystallization or cryo-EM.

**Preparation of promoter complex for structure determination.** The forked DNA template for crystallization and cryo-EM work was obtained by hybridization of two linear oligonucleotides given in Supplementary Table 5. The 18 base-long ss template contained the AR9 late promoter P077, while its 14 bp-long ds segment spanned positions from +3 to +16 relative to the TSS (Fig. 8b). Both oligonucleotides were synthesized by IDT with dual PAGE and HPLC purification at a final concentration of 100 μM each. They were annealed together in a buffer containing 20 mM Bis-tris propane pH 6.8, 100 mM NaCl, 4 mM MgCl$_2$, 0.5 mM EDTA, which was incubated at 65 °C for 1 min and then cooled down to 4 °C at a rate of 1 °C per minute. A 1.5-fold molar excess of the DNA template was added to the holoenzyme and incubated for 30 min at room temperature. The final concentrations of the reagents were as follows: 10 mg/mL for the protein (34 μM) and 50 μM for the DNA template. This specimen was used for crystallization and cryo-EM work directly.

**Crystallization of AR9 nvRNAP.** The initial crystallization screening was carried out by the sitting drop method in 96 well ARI Intelliwell-2 LR plates using Jena Bioscience crystallization screens at 19 °C. PHOENIX pipetting robot (Art Robbins Instruments, USA) was employed for preparing crystallization plates and setting up drops, each containing 200 nL of the protein and the same volume of well solution. Optimization of crystallization conditions was performed in 24 well VDX plates and thin siliconized cover slides (both from Hampton Research) by hanging drop vapor diffusion. The best crystals were obtained as follows: (i) a 1.5 μl aliquot of AR9 nvRNAP core (4.5 mg/mL) was mixed with an equal volume of a solution containing 100 mM Tricine pH 8.8, 270 mM KNO$_3$, 15 % PEG 6000, 5 mM MgCl$_2$, and incubated as a hanging drop over the same solution; (ii) a 1.5 μl aliquot of tagless AR9 nvRNAP core (7.5 mg/mL) was mixed with an equal volume of a solution containing 150 mM Malic acid pH 7, 150 mM NaCl, 14 % PEG 3350 and incubated as a hanging drop over the same solution; (iii) a 1.5 μl aliquot of the AR9 nvRNAP promoter complex (10 mg/mL) was mixed with an equal same volume of a solution containing 150 mM MIB pH 5, 150 mM LiCl, 13 % PEG 1500 and incubated as a hanging drop over the same solution. Some crystal reached their final size the next day and some grew for two weeks at 19 °C temperature.

**Preparation of heavy-atom derivative crystals.** The following compounds were tested for heavy-atom derivatization of AR9 nvRNAP core crystals (by co-crystallization and soaking): SrCl$_2$, GdCl$_3$, Na$_2$WO$_4$, HgCl$_2$, Pb(NO$_3$)$_2$, thimerosal (2-(C$_2$H$_5$HgS)C$_6$H$_4$CO$_2$Na), 10 compounds containing Eu and Yb atoms (JBS Lanthanide Phasing Kit), three compounds containing W (JBS Tungstate Cluster Kit) and one cluster compound containing Ta (Ta$_6$Br$_{12}$ JBS Tantalum Cluster Derivatization Kit). The crystals were soaked in a range of concentrations of heavy atom compounds (between 0.1 mM and 100 mM) that were added to the crystal-lization solution. The soaking time was varied from 2 h to 2 days. Among all examined conditions, only solutions containing 10 mM thimerosal or 1 mM tantalum bromide resulted in heavy-atom derivatization (judging by the presence of an anomalous signal in X-ray diffraction data) upon overnight soaking.

To produce a Se-methionine (SeMet) derivative of the AR9 nvRNAP core, the corresponding plasmid was transformed into B834(DE3) chemically competent E. coli cells. The cells were first grown in LB medium until the optical density OD$_{600}$ reached a value of 0.35. The cells were then pelleted by centrifugation at 4000 × $g$ for 10 min at 4 °C and transferred to the SelenoMet Medium (Molecular Dimensions) that was supplemented with ampicillin at a concentration of 100 μg/mL. The protein expression then proceeded according to the manufacturer's instructions. All the subsequent steps were the same as for the native protein.

**X-ray data collection and reduction.** Cryoprotectant solutions were prepared by replacing 25% of water in the crystallization solution (hanging drop well solution) with ethylene glycol, which was found to be the best cryoprotectant by trial and error. The crystals were either soaked for 1–5 min in the cryoprotectant solution or briefly dipped into it and then flash frozen in liquid nitrogen. Such frozen crystals were then transferred to a shipping Dewar and shipped to the APS (LS-CAT) or ALS (BCSB) synchrotrons for remote data collection. X-ray diffraction data and fluorescent spectra were collected in a nitrogen stream at 100 K. Heavy atom and SeMet derivative data were collected at the absorption peak wavelength (the white line, if present) of the X-ray fluorescence spectrum. The diffraction data were integrated and reduced using XDS[37].

**X-ray structure determination.** Initially, we aimed to solve the structure of the AR9 nvRNAP core by X-ray crystallography or cryo-EM and then use the core to solve the structure of the promoter complex. However, after nearly three year-long efforts, this approach was only partially successful. It resulted in an incomplete model of the core because large parts of it were disordered. The DNA template-bound promoter complex turned out to be better ordered than the core, and a nearly complete atomic model of the holoenzyme (as a component of the promoter complex) was built using cryo-EM data. This model was then used to complete the partially built X-ray-derived structure of the core, to interpret the cryo-EM map of the template-free holoenzyme, and to solve the crystal structure of the promoter complex.

None of the RNAP structures present in the PDB at the start of this project were sufficiently similar to solve the structure of the AR9 nvRNAP core by molecular replacement (MR), so crystallographic phases had to be obtained by a de novo phasing procedure (heavy atom isomorphous replacement or anomalous scattering). Severe anisotropy and inconsistent diffraction of AR9 nvRNAP core native crystals made this task extremely complicated and we had to screen hundreds of heavy-atom-soaked crystals for diffraction. The SeMet derivative diffracted to 5.5 Å resolution, which was insufficient to solve the Se substructure using anomalous scattering. Moreover, this derivative was not isomorphous to any of the native datasets.

An interpretable map was obtained by a convoluted procedure. A map in which the characteristic features of a DNA-dependent RNAP—two adjoining double-ψ β-barrel (DPBB) domains and several large α-helices, including the bridge helix (although split in the middle)—could be discerned (but without side-chain densities) was obtained by a multiple isomorphous replacement plus anomalous scattering method which was applied to the native, Ta$_6$Br$_{12}$, and thimerosal derivative datasets of the His-tagged AR9 nvRNAP core enzyme (see Supplementary Table 2). This map was calculated by the SHARP software package[38] which was run with mostly default settings. The most similar part of the archaeal RNAP structure[39] (PDB code 4ayb) identified using HHpred[40] was fitted into this density using Coot[41]. All parts that did not fit the density were removed, and all side chains were truncated at the Cβ atoms. The resulting model contained 832 alanine residues.

This model was then used to solve the structure of the Large unit cell thimerosal derivative dataset (Supplementary Table 2) by a MR procedure. This unique dataset was the result of the crystallization of a tagless version of the AR9 nvRNAP core (all protein chains had native N- and C-termini). It had a very large orthorhombic unit cell with eight molecules of the AR9 nvRNAP core in its asymmetric unit or about 17,800 amino acids (Supplementary Table 2). Remarkably, Phaser[42] was able to locate all eight copies of the AR9 nvRNAP core in this 3.8 Å resolution dataset using the 832-residue polyalanine fragment (obtained as described before) as a search model. This polyalanine model search model corresponded to about 1.8% of the total protein material in the asymmetric unit and 1% of the total asymmetric unit content if solvent atoms are taken into account. The density was then dramatically improved by 25 cycles of eightfold non-crystallographic symmetry (NCS) averaging using Parrot[43]. The resulting density was mostly continuous and

showed many bulky side chains, especially in the vicinity of the DPBB domains. Buccaneer[44] was then used for automatic model building into this map. The Buccaneer model was cleaned up manually and separate chain fragments were assembled into a new intermediate AR9 nvRNAP core model that contained 995 residues of which 937 had side chains. The new intermediate model was then used as a search model in a new round of MR by Phaser that was followed by NCS averaging using Parrot.

The new density was of sufficient quality to recognize the identity of many side chains and for manual model building using Coot. Structures of the *Mycobacterium tuberculosis* and *E. coli* RNAPs that became available in the meantime (PDB codes 5ZX3[45] and 6C9Y[46], respectively) were used to aid in chain tracing. In addition, the Large unit cell dataset was a thimerosal derivative that contained Hg atoms (identified with the help of anomalous Fourier synthesis) that were expected to bind to cysteine side chains. Thus, the Hg atoms were used as markers to maintain the chain register.

While the model of the nvRNAP core was being built using the electron density map of the Large unit cell tagless core enzyme (3.8 Å resolution, Supplementary Table 2), an interpretable cryo-EM map of the AR9 nvRNAP promoter complex was obtained (3.8 Å resolution, Supplementary Tables 3, 4). Despite having similar resolutions, the cryo-EM map was of better quality than the X-ray electron density, so the partial model of the core was transferred to the cryo-EM map of the promoter complex, and further rounds of the model building were performed using this cryo-EM map.

Some peripheral regions of the cryo-EM promoter complex map, namely, the β′ C gp154 TLID, residues 130-289 of βN gp105, and the peripheral parts of gp226, were too poor for reliable de novo model building. Fortuitously, we submitted the sequences of all five subunits comprising the AR9 nvRNAP holoenzyme to the CASP14 protein structure prediction competition. The DeepMind AlphaFold2 software[27] predicted the structure of all subunits, including the difficult-to-build domains, with excellent accuracy[47].

The atomic models of the AR9 nvRNAP core and holoenzyme that were derived from the cryo-EM data of the AR9 nvRNAP promoter complex were then used to solve the crystal structures of the Standard unit cell native AR9 nvRNAP core (3.3 Å resolution, Supplementary Table 2) and promoter complex (3.4 Å resolution, Supplementary Table 2), respectively, by MR using Phaser.

In addition to the crystallography software mentioned above, the CCP4 suite[48] was used in the data processing. The heavy atoms in the $Ta_6Br_{12}$ and thimerosal derivative datasets were found with the help of the SHELX suite of programs[49] controlled by the HKL2MAP[50] interface. In addition, Refmac5[51] was used for the refinement of atomic coordinates at various stages.

**Cryo-EM data acquisition of the AR9 nvRNAP promoter complex**. QUANTI-FOIL 1.2/1.3 copper grids were plasma cleaned for 30 s using the model 950 advanced plasma system by Gatan. 3 μL of 10 mg/mL of the AR9 nvRNAP holoenzyme (34 μM) and 50 μM of the DNA nucleotide in 20 mM Bis-tris propane pH 6.8, 100 mM NaCl, 2 mM $MgCl_2$, 0.5 mM EDTA was pipetted onto the grid and blotted using a Vitrobot (Thermo Fischer Scientific) at 100% humidity for 5 s. Following blotting, the sample was plunged into liquid ethane cooled by liquid nitrogen.

5351 (5760 × 4092 pixels) micrograph movies were collected using the EPU software on a Titan Krios 300 kV electron microscope with a BioQuantum K3 imaging filter with a 20-eV slit. Each movie contained 56 frames collected over 1.5 s, with a frame dose of 0.78 e/Å[2] and pixel size of 1.09 Å. Movies were collected over a defocus range of −1 to −4 μm (Supplementary Table 3).

**Cryo-EM image processing of the AR9 nvRNAP promoter complex**. Image processing was performed using Eman2[52], Relion3.0[53], and CryoSPARC3.0[54] (Supplementary Figs. 7, 8, Supplementary Table 3). All movies were motion corrected using MotionCor2[55]. Estimation of the contrast transfer function (CTF) parameters was performed by CTFFIND4.1[56] over the resolution range of 5.0–30.0 Å. E2boxer[52] was used for particle picking, resulting in 420,791 particles with a box size of 300 pixels. Particle box coordinates were used by Relion3.0 to extract the boxes. 2D classification by Relion3.0 resulted in 243,976 particles belonging to high-quality classes. These particles were imported into CryoS-PARC3.0, where ab initio reconstruction was carried out using three models. Following the ab initio reconstruction, 3D classification was performed with five classes, a box size of 150 pixels to improve speed, a batch size of 2000 particles per class and an assignment convergence criterion of 2%. Non-Uniform (NU) refinement[57] was executed using both the 106,867 particles and the map from the most populous class of 3D classification. 3D local refinement was then carried out using an alignment resolution of 0.25° and NU refinement. Subsequently, local CTF-refinement was performed using a search range of 3.5–20.0 Å. Following this, around round of NU refinement and 3D local refinement with an alignment resolution of 0.25° and NU-refinement was performed. The resulting map was sharpened with a B-factor of −138 Å[2].

**Cryo-EM data acquisition of the AR9 nvRNAP holoenzyme**. TED PELLA 200 mesh PELCO NetMesh copper grids were plasma cleaned for 30 s using the model 950 advanced plasma system by Gatan. 3 μl of 20 mg/ml His-tag 5 s nvRNAP in 20 mM Bis-tris propane pH 6.8, 100 mM NaCl, 4 mM $MgCl_2$, 0.5 mM EDTA buffer was pipetted onto the grid and blotted using a Vitrobot (Thermo Fischer Scientific) at 100% humidity for 5 s. Following blotting, the sample was plunged into liquid ethane cooled by liquid nitrogen.

2691 (3838 × 3710 pixels) micrograph movies were collected using the EPU software on a Titan Krios 300 kV electron microscope with a K2 Summit camera and a 20-eV slit. Each movie contained 40 frames collected over 8 s, with a frame dose of 1.08 e/Å[2] and pixel size of 1.08 Å (Supplementary Table 3). Movies were collected with a target defocus of −1.8 μm. Images were collected with a 30-degree tilt.

**Cryo-EM image processing of the AR9 nvRNAP holoenzyme**. Image processing was performed using Relion2.0[53]. All movies were motion corrected using MotionCor2[55]. Estimation of the contrast transfer function (CTF) parameters was performed by Gctf[58]. Particle picking resulted in 227,577 particles with a box size of 200 pixels. 2D and 3D classification by Relion2.0 resulted in 104,471 particles belonging to high-quality classes. The resulting particles were refined to a resolution of 4.4 Å (Supplementary Table 3). The resulting map was sharpened with a B-factor of −87 Å[2].

**Refinement of atomic models**. Refinement of crystallographic and cryo-EM models was performed using Phenix[59] and Coot[41] using reciprocal and real-space refinement protocols, as appropriate. Additional details describing the model building and the analysis of AlphaFold2 models are given elsewhere[47].

**Gp226 cloning, purification, and limited proteolysis**. The AR9 gene *226* was PCR amplified from AR9 genomic DNA and cloned into the pQE-2 vector (QIAGEN) between the SacI and SalI restriction sites. The resulting plasmid was transformed into BL21 (DE3) chemically competent *E. coli* cells. The culture (7 L) was grown at 37 °C to an OD$_{600}$ of 0.5 in LB medium supplemented with ampicillin at a concentration of 100 μg/mL, and recombinant protein overexpression was induced with 1 mM IPTG for 4 h at 22 °C. Cells containing over-expressed recombinant protein were harvested by centrifugation and disrupted by sonication in buffer C followed by centrifugation at 15,000 × *g* for 30 min. Cleared lysate was loaded on a 5 mL Ni-NTA column (Qiagen) equilibrated with buffer C, washed with 5 column volumes of buffer C and with 5 column volumes of buffer C containing 20 mM Imidazole. Then, elution with buffer C containing 200 mM Imidazole was carried out. Fractions containing gp226 were pooled, concentrated and subjected to gel-filtration on a Superdex 200 10/300 (GE Healthcare) column equilibrated with buffer C. The fractions containing gp226 monomer were concentrated to a final concentration of 1 mg/mL and used for the limited proteolysis experiment.

Trypsin digestion of gp226, which had a concentration of 80 ng/μL, was carried out in 20 ul of the digestion buffer (50 mM $NaH_2PO_4$ pH 8.0, 300 mM NaCl) that contained a range of trypsin concentrations (Sigma-Aldrich). The trypsin to gp226 molar ratios were from 0.03 to 0.6. The reactions were allowed to proceed for 1 h at 25 °C and stopped by the addition of Laemmli loading buffer and immediate boiling. The reaction products were analyzed by denaturing SDS polyacrylamide gel electrophoresis (SDS-PAGE) with subsequent mass-spectrometry according to a standard protocol[1] (Supplementary Data 1).

**DNA templates for transcription assay**. Long DNA templates containing late AR9 promoters were prepared by polymerase chain reaction (PCR). PCRs were done with Encyclo DNA polymerase (Evrogen, Moscow) and the AR9 genomic DNA as a template, with a standard concentration of dNTPs (Thermo Fisher Scientific) to obtain DNA fragments with thymine or in the presence of dUTP (Thermo Fisher Scientific) in place of dTTP to obtain DNA fragments with uracil. Oligonucleotide primers used for PCR are listed in (Supplementary Table 5).

Short double-stranded and partially single-stranded DNA templates containing the P077 promoter with uracils and thymines at certain positions were prepared by annealing of oligonucleotides ordered from Evrogen (Moscow) and listed in (Supplementary Table 5). To prepare specific DNA templates, two corresponding oligonucleotides were annealed together by mixing in a buffer containing 40 mM Tris-HCl pH 8, 10 mM $MgCl_2$, and 0.5 mM DTT, incubating at 75 °C for 1 min and cooling down to 4 °C by a decrement of 1 °C per minute.

**In vitro transcription**. Multiple-round run-off transcription reactions were performed in 5 μL of transcription buffer (40 mM Tris-HCl pH 8, 10 mM $MgCl_2$, 0.5 mM DTT, 100 μg/mL bovine serum albumin (Thermo Fisher Scientific), and 1 U/μL RiboLock RNase Inhibitor (Thermo Fisher Scientific)) and contained 100 nM AR9 nvRNAP holoenzyme and 100 nM DNA template. The reactions were incubated for 10 min at 37 °C, followed by the addition of 100 μM each of ATP, CTP, and GTP, 10 μM UTP and 3 μCi [α-$^{32}$P]UTP (3000 Ci/mmol) (Figs. 1c, 8b) or 100 μM each of ATP, UTP, GTP, 10 μM CTP and 3 μCi [α-$^{32}$P]CTP (3000 Ci/mmol) (Fig. 8d, f). Reactions proceeded for 30 min at 37 °C and were terminated by the addition of an equal volume of denaturing loading buffer (95% formamide, 18 mM EDTA, 0.25% SDS, 0.025% xylene cyanol, 0.025% bromophenol blue). The reaction products were resolved by electrophoresis on 6–23% (w/v) polyacrylamide

gel containing 8 M urea. The results were visualized with a Typhoon FLA 9500 scanner (GE Healthcare) (Supplementary Data 1).

**Molecular dynamics general methods.** Simulations were carried out on both the LS5 and Stampede2 systems at the Texas Advanced Computing Center (TACC) using NAMD 2.10[36]. The CHARMM36 force field was used[60]. Production runs were performed in the isothermal isobaric (NPT) ensemble using Langevin dynamics and a Langevin piston[61]. Alchemical transformations were analyzed by the ParseFEP package[62]. Entropic restraints were calculated via thermodynamic integration. Collective variables were implemented via the colvars module in NAMD[63]. The energy of non-bonded VdW interactions for distances exceeding 10 Å was smoothly decreased to equal zero at 12 Å. A 2 fs time step was used in all simulations. Long-range electrostatics was calculated with the help of the Particle Mesh Ewald algorithm[64]. During alchemical transformation, a soft core VdW radius of 4 $^2$ was used to improve convergence and accuracy[65]. Both alchemical and restraint calculation simulations were carried out bidirectionally. The Bennett Acceptance Ratio maximum likelihood estimate[66] was used to determine free energy change for alchemical transformations. The double decoupling method (DDM) was implemented as described[35].

**Molecular dynamics system setup.** The protein structure description files for both structures—the AR9 nvRNAP holoenzyme in complex with the 3′-−11UUG−9-5′ oligonucleotide bound to the promoter binding pocket and for the 3′-−11UUG−9-5′ oligonucleotide in the promoter bound conformation—were generated using the psfgen plugin of VMD[67]. Solvation was performed using TIP3 water[68] in a box with 15 Å padding in each direction and ionized in 0.1 M NaCl. The holoenzyme-DNA and DNA systems were enclosed in periodic boxes with cell dimensions of (176, 147, 133 Å) and (42, 42, 41 Å), respectively, and contained 91,177 and 2166 water molecules, respectively. Both systems were first minimized for 1,000,000 steps while restraints and constraints on the protein (in the holoenzyme system), DNA, and water atoms were gradually removed. Both systems were heated from 0 to 300 K in 5 K increments for a total of 19.2 ns with constraints on backbone atoms. The holoenzyme-DNA and DNA systems were then equilibrated with minimal constraints in the isothermal-isobaric (NPT) ensemble for 100 and 50 ns, respectively. The parameters describing the MD setup and resulting atomic models are organized into folders and files with self-explanatory names and are given in Supplementary Data 3.

**Definition of collective variables.** For the implementation of the DDM method via alchemical transformations, the system must be (harmonically) constrained such that the finite sampling can be focused on relevant regions of phase space. The entropic cost of applying these restraints is evaluated in separate independent simulations. The phase space and the entropic cost are connected to each other through a set of collective variables that are applied to atoms during simulations.

Only one collective variable is needed to restrain the conformation of the oligonucleotide in bulk water (the bulk water DNA system): the RMSD of all non-H DNA atoms relative to the equilibrated state. In the holoenzyme-DNA complex, seven collective variables are required—six to define the orientation and position of the rigid DNA molecule relative to the holoenzyme complex and one to define the conformation of the DNA. Similar to the bulk water DNA case, the RMSD of all non-H DNA atoms relative to the equilibrated state is used as the collective variable to restrain the conformation of DNA. We characterize the orientation of the rigid DNA molecule via the relative position of the backbone atoms of −11UUG−9 to those of gp226 V206, gp105 N382, and gp226 K262. Accordingly, the orientation of DNA relative to the holoenzyme is characterized by six collective variables: $r$—the distance between −11U and gp226 V206); $\phi$—the angle between gp226 V206, −11U, and −9G); $\chi$—the angle between −11U, gp226 V206, and gp105 N382); $\theta$—the dihedral angle between gp226 V206, −11U, −9G, and −10U); $\psi$—the dihedral angle between gp105 N382, gp226 V206, −11U, and −9G); $\xi$—the dihedral angle between −11U, gp226 V206, gp105 N382, and gp226 K262).

The harmonic force constraint constants applied to the distance-type (RMSD and $r$) and angular collective variables were 10 kcal/mol/Å$^2$ and 0.1 kcal/mol/deg$^2$, respectively. The equilibrium positions for all harmonic restraints were derived from the equilibrated holoenzyme-DNA structure. For restraint estimation simulations, harmonic forces were varied smoothly using a target force exponent value of 4.0. The lambda schedule focused near the value 1.0 to improve simulation convergence and ensure thermodynamic micro-reversibility: [1.00, 0.999, 0.99, 0.95, 0.90, 0.85, 0.80, 0.75, 0.70, 0.65, 0.60, 0.55, 0.50, 0.45, 0.40, 0.35, 0.30, 0.20, 0.10, 0.00]. The reverse sequence was used for the backward simulation.

**Thermodynamic cycle.** The standard binding free energy was calculated by combining the results of four separate simulations which represent the four vertical reactions of the thermodynamic cycle (Fig. 9a). These simulations evaluate the following parameters (Supplementary Table 6): (1) the entropic cost of restraining the promoter DNA to the "bound" state in the promoter binding pocket by adding/removing conformational restraints on the promoter DNA ($\Delta G_{restrain}^{bound}$); (2) the free energy of coupling/decoupling the promoter DNA from the binding pocket via alchemical transformations with restraints on the conformation of the promoter DNA ($\Delta G_{alchemical}^{bound}$); (3) the entropic cost of restraining the promoter DNA to the

bound conformation in bulk water by adding/removing conformational restraints on the promoter DNA ($\Delta G_{restrain}^{bulk\ water}$); (4) the free energy of coupling/decoupling the promoter DNA from bulk water via alchemical transformations with restraints on the conformation of the promoter DNA ($\Delta G_{alchemical}^{bulk\ water}$). The change in free energy resulting from these transitions is then calculated via thermodynamic integration[69] and free energy perturbation[70] methods. The results were validated by checking for micro-reversibility and the absence of hysteresis[62,66]. Following the completion of the thermodynamic cycle, the standard binding free energy of promoter DNA to the holoenzyme binding pocket was found to be −6.9 ± 2.8 kcal/mol.

**Molecular dynamics error analysis.** An upper bound on the error of $\Delta G_{restrain}^{bound}$, $\Delta G_{restrain}^{bulk\ water}$ and $\Delta G_{alchemical}^{bulk\ water}$ was determined by the hysteresis between backward and forward simulations[35]. The error in $\Delta G_{alchemical}^{bound}$ was determined by performing 3 replicates of the simulation and evaluating the standard deviation (Supplementary Table 6).

**Software used in figure preparation and structure comparison.** Molecular structure figures (Figs. 2, 3, 4, 5, 6, 7, 8a, c, e, f, 9a; Suppl. Figs. 2, 4a, 6, 7d, and 8d) were made using ChimeraX[71]. The local resolution maps shown in Suppl Figs. 7d and 8d were calculated using ResMap[72]. The electrostatic potential surface distributions shown in Fig. 8e, f, Suppl. Fig. 6 were calculated using APBS[73]. The structure-based sequence alignment shown in Supplementary Fig. 3 was calculated using Chimera[74]. Supplementary Fig. 5 was created using Esprit[75], Blast[76], ClustalX[77]. All RMSDs are calculated using the SSM superposition routine of Coot[41].

**Reporting summary.** Further information on research design is available in the Nature Research Reporting Summary linked to this article.

## Data availability

All macromolecular structure data generated in this study have been deposited to the Protein Data Bank and Electron Microscopy Data Bank under the following accession numbers: PDB code 7S00 (AR9 nvRNAP core X-ray structure); PDB code 7S01 (AR9 nvRNAP promoter complex X-ray structure); PDB code 7UM0 (AR9 nvRNAP promoter complex cryo-EM structure); PDB code 7UM1 (AR9 nvRNAP holoenzyme cryo-EM structure); EMDB code EMD-24763 (AR9 nvRNAP promoter complex cryo-EM density); EMDB code EMD-24765 (AR9 nvRNAP holoenzyme cryo-EM density).

The unedited images of the in vitro transcription assays and polyacrylamide gel photographs can be found in Supplementary Data 1. The coordinates of the phiKZ gp68 model created by AlphaFold Colab are in Supplementary Data 2. The molecular dynamics setup files and coordinates are in Supplementary Data 3.

Publicly available protein atomic models with the following PDB codes were used in the study: 4AYB (Archaeal RNA Polymerase)[39], 5ZX3 (Mycobacterium tuberculosis RNA polymerase holoenzyme with ECF sigma factor sigma H)[45], 6C9Y (Cryo-EM structure of E. coli RNAP sigma70 holoenzyme)[46], 5IPM (SigmaS-transcription initiation complex with 4-nt nascent RNA)[21], 6JBQ (CryoEM structure of Escherichia coli sigmaE transcription initiation complex containing 5nt of RNA)[22], and 7OGP (Structure of the apo-state of the bacteriophage PhiKZ non-virion RNA polymerase)[8].

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

## Acknowledgements

This work was supported by the Skoltech NGP Program (Skoltech-MIT joint project), the Russian Science Foundation (Grant 19-74-00011 to M.L.S.), the Russian Foundation for Basic Research (Grant 20-34-90079 to J.V.G.), the National Institutes of Health (Grant R01GM130942/Subaward 0518GWB837 to S.B.), the Busch Biomedical Grant Program (K.V.S.), and the Ministry of Science and Higher Education of Russian Federation (Agreement No. 075-10-2021-114 to K.V.S.). The work was also supported by the UTMB Department of Biochemistry and Molecular Biology (A.F., M.L.S., and P.G.L.) and by the UTMB Sealy Center for Structural Biology and Molecular Biophysics (P.G.L.). The MD work was performed using the computing facilities of the Texas Advanced Computing Center (TACC, http://www.tacc.utexas.edu) at The University of Texas for which we are very grateful. We thank the Stanford-SLAC Cryo-EM Facilities, supported by Stanford University, SLAC, and the National Institutes of Health S10 Instrumentation Programs that were used to collect the AR9 nvRNAP holoenzyme cryo-EM data. We acknowledge the use of the Advanced Photon Source, a U.S. Department of Energy (DOE) Office of Science User Facility operated for the DOE Office of Science by Argonne National Laboratory under Contract No. DE-AC02-06CH11357. We thank the staff of the LS-CAT Sector 21 beamlines that are supported by the Michigan Economic Development Corporation and the Michigan Technology Tri-Corridor (Grant 085P1000817). We acknowledge the use of the Berkeley Center for Structural Biology (supported in part by the Howard Hughes Medical Institute) at the Advanced Light Source (a Department of Energy Office of Science User Facility under Contract No. DE-AC02-05CH11231) and we thank the staff of the beamline 5.0.2. Finally, we thank Dr. Mark A. White for his help and assistance with the initial crystallization and X-ray data collection of the AR9 nvRNAP core, Dr. Michael B. Sherman for his help with the cryo-EM data collection, and Dr. Tatyana O. Artamonova for mass-spectrometry analysis of gp226 digestion products. The research reported in this paper extensively used the facilities and resources of the UTMB SCSB Macromolecular Structure X-ray Laboratory and UTMB SCSB Cryo-EM Laboratory. We are grateful to Andriy Kryshtafovych, one of the Critical Assessment of Protein Structure Prediction 14 (CASP 14) competition organizers, and the AlphaFold Team for sharing their predicted models of all five subunits of the AR9 nvRNAP holoenzyme prior to the conclusion of the CASP 14 competition. These models were used in model building and validation of chain tracing of the AR9 nvRNAP holoenzyme. In addition, the atomic model of phage phiKZ gp68 was created with the help of AlphaFold Colab. The full list of the AlphaFold Team members is given in the Supplementary Information with John Jumper representing the entire AlphaFold Team in the author list.

## Author contributions

K.V.S. and M.L.S. conceived the study. M.L.S. cloned, purified, and crystallized AR9 nvRNAP core, tagless AR9 nvRNAP core, and AR9 nvRNAP holoenzyme in complex with promoter DNA, derivatized crystals, prepared samples for cryo-EM, purified gp226 and performed limited trypsinolisis. A.F. obtained and analyzed all cryo-EM reconstructions, built parts of atomic models, and performed all MD work. A.V.D. purified AR9 nvRNAP holoenzyme and its mutants and performed in vitro transcription assays under the supervision of M.L.S. J.G. under the supervision of M.L.S. crystallized the AR9 nvRNAP holoenzyme in complex with promoter DNA. P.G.L. collected X-ray data, solved all crystal structures, and built and refined all atomic models. J.J. and the A.F. team created models of all five AR9 nvRNAP holoenzyme proteins that were used by P.G.L. and A.F. in the interpretation of cryo-EM and X-ray crystallography electron density maps. J.J. created the model of phiKZ gp68. A.F., M.L.S., S.B., and P.G.L. analyzed the structures. P.G.L. and A.F. wrote the manuscript, which was read, edited, and approved by all authors.

## Competing interests

The authors declare no competing interests.
