## [Peer Review File · Nature Communications]

REVIEWER COMMENTS

Reviewer #1 (Remarks to the Author):

The paper from Fraser et al. reports on a set of new structures of the AR9 non-virion RNA polymerase, nvRNAP. This polymerase has a unique promoter recognition mechanism, which is resolved here.

I found the combined set of structures, experiments, and simulations quite impressive overall. I can specifically comment on the simulation work. The free-energy calculations were carefully executed. The thermodynamic cycle appears to be correct.

Something that detracts slightly from the author's approach is that they used only a tri-nucleotide instead of a longer piece of DNA. I imagine that interactions between the three nucleotides and neighboring ones would contribute differently in the bound and unbound states. I'm not sure it's worth re-doing the calculations, as I don't think it would be large enough to change the conclusion, but the authors should comment on this.

Reviewer #2 (Remarks to the Author):

The manuscript reports the structures of bacteriophage AR9 non-virion RNA polymerase (nvRNAP) core enzyme, holoenzyme, and promoter complex. The structures reveal the architecture of a non-canonical multi-subunit RNA polymerase and the mechanism by which a multi-subunit RNA polymerase recognizes the uridine of template strand promoter DNA. The work sheds lights on the evolution of multi-subunit RNA polymerase and will be of intense interest to the transcription field. While the data is solid, the manuscript can be further improved by addressing the following questions.

1. Previous studies have shown that the clamp of multi-subunit RNAP undergoes swing motions that open the active center cleft to allow entry of the nucleic acid scaffold during initiation or that close the cleft around the nucleic acid scaffold to enable processive elongation. Compared to the cryo-EM structure of nvRNAP holoenzyme, does the clamp change its conformation in the cryo-EM structure of promoter complex?

2. Page 8, line 148, the statement “it carries a strong negative charge on its DNA-facing surface suggesting that it interacts with the downstream dsDNA in a sequence independent manner” is problematic. DNA doesn’t usually interact with negatively charged protein surface. According to figure 2a, the TL insertion domain probably doesn’t interact with DNA at all. Considering that the TL insertion domain is negatively charged and located in the main channel of nvRNAP core enzyme as $\sigma 1.1$ of $\sigma 70$, is it possible that the TL insertion domain functions as a mimic of DNA and reduces nonspecific RNAP-DNA interactions?

3. Page 11, line 212, a supplementary figure of sequence alignment is needed to show the sequence conservation of the 2.1bis helix.

4. Figure 2b, the sequence of the non-template strand single-stranded DNA is wrong.

5. Figure 8a, some interacting residues mentioned in the main text (e.g. gp226 residues I207 and Y210) are invisible.

6. Figure 8c, please change S247 to S245.

7. Please add a figure of cryo-EM image processing workflow for AR9 nvRNAP holoenzyme in the supplementary file.

Reviewer #3 (Remarks to the Author):

Fraser, Sokolova, et al. present the structural basis of promoter recognition and melting by the AR9 nvRNAP. Using a combination of structural methods and functional assays, they present important insights into how a polymerase similar in overall architecture to the bacterial RNAP can interact with its promoter very differently. They also provide the basis of how this mechanism is tuned to a uracil-containing DNA genome. The work represents a wealth of biophysical and biochemical data and appreciably hard work. However, some issues require addressing before publication (which I do recommend once addressed):

Major-

1) It was difficult to follow the pipeline as described in the methods. Considering the various techniques and structures, the authors should include in the main figures a visual pipeline of the structures with the resolutions, the overall map in transparent white or grey, and names that they can use when referring to which structures they used to discuss or compare the various features. In addition, a table should be added at the beginning of the paper with this info. i.e., they can name the structure nvRNAP-RPoEM to indicate the structure has all components of an open complex (initiation factor and NA) or nvRNAP-holoXR to indicate Xray, etc. The table should contain the following info: Name of the structure, method used, resolution, features present, PDB ID/mapID. The models should then be referenced in the figure legends to indicate which structures went into the figure. The table/pipeline will guide the readers to the resolution and what was observed in the structures. Right now, it's very confusing, and the readers would have a hard time knowing which PDB to refer to for specific features.

2) The authors resolved two cryo-EM maps but did not deposit models yet used the models to formulate a hypothesis on promoter melting. They should dock the observable parts of the RNAP, do a rigid refinement of the domains, and deposit the PDBs if they wish to refer to the models derived from the cryoEM studies in this paper. This would be relatively easy to do since they have the models but chose not to deposit them.

3) Along this line, Figure 1 has a structure (one of the X-ray structures of the promoter complex, I assume) used for most of the analysis. An overall map should be shown here so the readers can evaluate the density of the map to the model.

4) Figure 7c. The CryoEM map is very low resolution (contoured to 2), and the crucial pseudo -35 element has poor density. Therefore, the local resolution map needs to be calculated and presented here.

5) Figure 8b (full gel in S5)-There are multiple bands-some as strong as the "main bands." Please explain the expected product size and the presence of additional bands.

6) Figure 8 represents some of the more significant findings of the paper, but there is no figure illustrating the density map of the nucleic acids. A mesh map with a view illustrating the residues and bases that are discussed/mutated) should be shown here or in the supplement. This is absolutely essential.

7) A discussion of phikZ should precede the results. The authors casually compare parts of their structure to the structure of this nvRNAP. I think they can state a holoenzyme of another large nvRNAP structure was previously determined. They can list the differences (uracil, host, etc.) and the distinguishing characteristics of this paper (promoter recognition and melting) at the beginning.

8) They should also include a sequence alignment of AR9 RNAP to phikZ and a bacterial RNAP and highlight the various features described in figures 2 and 3.

9) The authors state that direct recognition of the template strand shifts the existing paradigm of promoter recognition by holo-RNAP. This promoter does not have -35 and specific interaction with sigma that is different (no sequence and low structural homology in "reg 4"). Therefore, it is unfair to compare promoter recognition in bacteria and AR9 since the "sigma" differs and promoter compositions are different. I would ask them to rephrase line 23 in the abstract "shift this paradigm" to state that the mechanism differs from the bacterial system.

Minor-

1) Line 78- Maps determined by cryo-EM are not electron density maps (they are cryo-EM maps or coulombic potential maps).

2) Line 80 "has been" suggested a previous study- use "was" to indicate this study.

3) Lines 148-149. Could the insertion domain serve a similar role as sigma region 1 (prevent nonspecific DNA binding)? If it's negatively charged, why do the authors assume it interacts with DNA- I would think the opposite?

4) Lines 190-191: region 4 is just as conserved structurally as those sigmaD2 and finger, so the statement is incorrect.

5) Lines 240-241. I think the authors should mention that the interactions likely facilitate stabilization of bubble and unwinding (not just DNA recognition).

6) Figures can have better labels (Indicate in the figures which are bacterial, AR9, Xray, or Cryo-EM) in figures 2-8.

Our answers to the questions and comments below, as well as all modifications to the text, figure legends, and Supplementary Information file are in blue font.

Dear Reviewers,

Thank you very much for your time and effort spent on reading, understanding, and correcting our manuscript. Your work is greatly appreciated.

REVIEWER COMMENTS

Reviewer #1 (Remarks to the Author):

The paper from Fraser et al. reports on a set of new structures of the AR9 non-virion RNA polymerase, nvRNAP. This polymerase has a unique promoter recognition mechanism, which is resolved here.

I found the combined set of structures, experiments, and simulations quite impressive overall. I can specifically comment on the simulation work. The free-energy calculations were carefully executed. The thermodynamic cycle appears to be correct.

Thank you!

Something that detracts slightly from the author's approach is that they used only a tri-nucleotide instead of a longer piece of DNA. I imagine that interactions between the three nucleotides and neighboring ones would contribute differently in the bound and unbound states. I'm not sure it's worth re-doing the calculations, as I don't think it would be large enough to change the conclusion, but the authors should comment on this.

The DDM method estimates the energetics of creating a negatively charged promoter DNA in aqua and in the promoter binding pocket. A substantial component to this process is the energetics of solvating a charged molecule (this is likely > 90% of the total energetics). For this reason, the error in our simulations is dominated by the error in the estimation of the solvation energy. By including a 4th nucleotide, we would substantially increase the error. Furthermore, there is more conformational entropy in the 4th nucleotide than in the first three because it is disordered in the cryo-EM structure. Given that the DDM requires explicit restraints to applied on the entire nucleotide, the estimation of this entropic term lies beyond the scope of the DDM method.

Reviewer #2 (Remarks to the Author):

The manuscript reports the structures of bacteriophage AR9 non-virion RNA polymerase (nvRNAP) core enzyme, holoenzyme, and promoter complex. The structures reveal the architecture of a non-canonical multi-subunit RNA polymerase and the mechanism by which a multi-subunit RNA polymerase recognizes the uridine of template strand promoter DNA. The work sheds lights on the evolution of multi-subunit RNA polymerase and will be of intense interest to the transcription field. While the data is solid, the manuscript can be further improved by addressing the following questions.

1. Previous studies have shown that the clamp of multi-subunit RNAP undergoes swing motions that open the active center cleft to allow entry of the nucleic acid scaffold during initiation or that close the cleft around the nucleic acid scaffold to enable processive elongation. Compared to the cryo-EM structure of nvRNAP holoenzyme, does the clamp change its conformation in the cryo-EM structure of promoter complex?

Thank you for your question. It prompted us to create and refine an atomic model of the AR9 nvRNAP holoenzyme. We found that the clamp closes upon binding to the promoter, but the promoter-bound and promoter-free structures are very similar. The clamp rotates by only 3 degrees. Fig. 5 shows how well the promoter-bound structure fits – as a rigid body – into the cryo-EM map of the holoenzyme. We now describe the parameters of this fit and the comparison of the two conformations in the revised text (bottom of page 8).

2. Page 8, line 148, the statement “it carries a strong negative charge on its DNA-facing surface suggesting that it interacts with the downstream dsDNA in a sequence independent manner” is problematic. DNA doesn’t usually interact with negatively charged protein surface. According to figure 2a, the TL insertion domain probably doesn’t interact with DNA at all. Considering that the TL insertion domain is negatively charged and located in the main channel of nvRNAP core enzyme as $\sigma^{1.1}$ of σ^{70} , is it possible that the TL insertion domain functions as a mimic of DNA and reduces nonspecific RNAP-DNA interactions?

We agree. Although electrostatic repulsion is a bona fide “interaction”. We added your suggestion to the revised text (middle/upper part of page 8).

3. Page 11, line 212, a supplementary figure of sequence alignment is needed to show the sequence conservation of the 2.1bis helix.

The full sequence alignment is shown in Supplementary Fig. 4. A WebLogo of the 2.1bis helix is shown as an inset in Fig. 6a.

4. Figure 2b, the sequence of the non-template strand single-stranded DNA is wrong.

Thank you very much for catching this error! Corrected.

5. Figure 8a, some interacting residues mentioned in the main text (e.g. gp226 residues I207 and Y210) are invisible.

The residues are visible and labeled in the previous figure (in Fig. 7a, 7b).

6. Figure 8c, please change S247 to S245.

Again – thank you very much! Corrected.

7. Please add a figure of cryo-EM image processing workflow for AR9 nvRNAP holoenzyme in the supplementary file.

New Supplementary Fig. 7.

Reviewer #3 (Remarks to the Author):

Fraser, Sokolova, et al. present the structural basis of promoter recognition and melting by the AR9 nvRNAP. Using a combination of structural methods and functional assays, they present important insights into how a polymerase similar in overall architecture to the bacterial RNAP can interact with its promoter very differently. They also provide the basis of how this mechanism is

tuned to a uracil-containing DNA genome. The work represents a wealth of biophysical and biochemical data and appreciably hard work. However, some issues require addressing before publication (which I do recommend once addressed):

Major-

1) It was difficult to follow the pipeline as described in the methods. Considering the various techniques and structures, the authors should include in the main figures a visual pipeline of the structures with the resolutions, the overall map in transparent white or grey, and names that they can use when referring to which structures they used to discuss or compare the various features. In addition, a table should be added at the beginning of the paper with this info. i.e., they can name the structure nvRNAP-RPoEM to indicate the structure has all components of an open complex (initiation factor and NA) or nvRNAP-holoXR to indicate Xray, etc. The table should contain the following info: Name of the structure, method used, resolution, features present, PDB ID/mapID. The models should then be referenced in the figure legends to indicate which structures went into the figure. The table/pipeline will guide the readers to the resolution and what was observed in the structures. Right now, it's very confusing, and the readers would have a hard time knowing which PDB to refer to for specific features.

Thank you for helping to make our paper more readable. Unfortunately, we cannot add another main text figure without removing one of the existing figures. We created a table – Supplementary Table 1 – that contains all the requested information. We introduced the requested abbreviations (e.g. AR9 nvRNAP-Pro-Xray). We added the abbreviated labels to figure legends and figure panels where a confusion might arise.

We do not think that we can call our promoter DNA-containing complex RPo because it does not contain a real transcription bubble.

2) The authors resolved two cryo-EM maps but did not deposit models yet used the models to formulate a hypothesis on promoter melting. They should dock the observable parts of the RNAP, do a rigid refinement of the domains, and deposit the PDBs if they wish to refer to the models derived from the cryoEM studies in this paper. This would be relatively easy to do since they have the models but chose not to deposit them.

The promoter complex cryo-EM-derived model is very similar to the crystal structure albeit it has lower resolution and contains a small fraction of DNA (see line 93 in the previous version of the MS). We reasoned that 1) we are not gaining any new information from this atomic model; 2) by depositing it to PDB we are reducing the accuracy of the entire database because a higher-resolution and better-quality model of the same specimen would also be deposited. For the holoenzyme structure, we do not think that an atomic model is required to tell that a domain is disordered (see Fig. 5).

In any case, we have refined and deposited atomic models of the promoter complex and holoenzyme to the PDB (see Supplementary Table 1 and 4).

3) Along this line, Figure 1 has a structure (one of the X-ray structures of the promoter complex, I assume) used for most of the analysis. An overall map should be shown here so the readers can evaluate the density of the map to the model.

Our attempts at finding a comprehensible and representative visualization of the map-to-model fit for the whole structure failed. The quality of the map decreases towards its periphery and this is what one sees in an overall view. More central and better defined parts of the model are obscured by the peripheral parts. We hope that Fig. 7 shows the most functionally relevant and important regions of the electron density and electrostatic potential distribution (e.g. cryo-EM “map”) in sufficient detail and as unbiased as possible for the AR9 nvRNAP-Pro-Xray and AR9 nvRNAP-Pro-cryoEM datasets – a composite omit map at 1.5 sigmas and the cryo-EM map at 5 sigmas,

respectively. The refinement statistics are shown in Supplementary Tables 2 and 4 and both, X-ray- and cryo-EM-derived models are of good quality.

4) Figure 7c. The CryoEM map is very low resolution (contoured to 2), and the crucial pseudo -35 element has poor density. Therefore, the local resolution map needs to be calculated and presented here.

We apologize for the lack of clarity in the figure legend (this has been fixed in the new version). The structure of the upstream oligo is copied from the X-ray promoter structure (AR9 nvRNAP-Pro-Xray) to demonstrate that it is present in the cryo-EM dataset but is disordered. It is not present in the refined cryo-EM model deposited to PDB as the resolution of the map is too low for model building. The purpose of this exercise (rendering the cryo-EM map at such a low contour level) is to show that the upstream oligo is present in the cryo-EM map, that its position is similar to its location in the X-ray dataset, and that this oligo can reach the CTD of gp226 on its own. It does not need a crystal lattice to do so.

Of note, as the quality of our data do not allow us to interpret the interaction of the upstream oligo with the pseudo -35 element binding motif in atomic detail, we can only manipulate the surface of the domain as a whole (see Fig. 8e, 8f, 8g). No local resolution map will make this interaction more definitive or help with the interpretation.

5) Figure 8b (full gel in S5)-There are multiple bands-some as strong as the “main bands.” Please explain the expected product size and the presence of additional bands.

The updated Supplementary Fig. 8 shows autoradiograms and photographs of all gels used in transcription assays. In the autoradiograms, the products of abortive and non-specific transcription that migrate faster and slower, respectively, than the promoter-specific transcription product of the expected size are labeled. Bands corresponding to the radioactive nucleoside triphosphate are also indicated. The approximate size of a transcription product can be derived from the mobility of dyes visible in the accompanying photograph of the gel. In a 23% polyacrylamide denaturing gel, xylene cyanol and bromophenol blue migrate faster than ~28 base- and ~8 base-long oligonucleotides, respectively. In a 16% polyacrylamide denaturing gel, xylene cyanol and bromophenol blue migrate as ~40 base- and ~10 base-long oligonucleotides, respectively.

6) Figure 8 represents some of the more significant findings of the paper, but there is no figure illustrating the density map of the nucleic acids. A mesh map with a view illustrating the residues and bases that are discussed/mutated) should be shown here or in the supplement. This is absolutely essential.

Indeed. Apologies for this oversight. See new Fig. 7c

7) A discussion of phiKZ should precede the results. The authors casually compare parts of their structure to the structure of this nvRNAP. I think they can state a holoenzyme of another large nvRNAP structure was previously determined. They can list the differences (uracil, host, etc.) and the distinguishing characteristics of this paper (promoter recognition and melting) at the beginning.

We briefly describe the phiKZ nvRNAP structure in the Intro (second paragraph) and the questions that remain. In fact, bioinformatics very convincingly showed that all such enzymes are related to cellular RNAPs (see, e.g. Ref. 8 - Viruses. 2021 Jan; 13(1): 63.). The paper about the structure of the phiKZ nvRNAP is thus confirmatory in its nature. It does not provide any new insight into the function of these enzymes. Of note, the absence of the rudder was not discussed in that paper at all.

8) They should also include a sequence alignment of AR9 RNAP to phikZ and a bacterial RNAP and highlight the various features described in figures 2 and 3.

The AR9 and phiKZ nvRNAP are as different from each other sequence-wise as each of them is from bacterial enzymes. These sequences cannot be aligned to each other without secondary structure information and HMM profiles. A BLAST search with either of these sequences does not hit bacterial or any other “canonical” RNAPs.

As requested, we created a figure (Supplementary Fig. 2) that shows a sparsely annotated, structure-based sequence alignment of AR9 nvRNAP, phiKZ nvRNAP and E. coli RNAP. The alignment was calculated automatically by UCSF Chimera. As the alignment is very poor and in some regions the phage enzymes are more similar to the E. coli enzyme than they are to each other, it appears to be prudent to compare either RNAP to the better-studied bacterial RNAP before a cohesive picture that unites these enzymes will appear.

9) The authors state that direct recognition of the template strand shifts the existing paradigm of promoter recognition by holo-RNAP. This promoter does not have -35 and specific interaction with sigma that is different (no sequence and low structural homology in “reg 4”). Therefore, it is unfair to compare promoter recognition in bacteria and AR9 since the “sigma” differs and promoter compositions are different. I would ask them to rephrase line 23 in the abstract “shift this paradigm” to state that the mechanism differs from the bacterial system.

Fair. Removed and reworded.

Minor-

1) Line 78- Maps determined by cryo-EM are not electron density maps (they are cryo-EM maps or coulombic potential maps).

The word “map” is almost as bad as cryoEM electron “density”. It is a jargon that we all have become accustomed to. Cryo-EM “maps” are scalar fields of electrostatic potential distribution. We have been trying to avoid using the word “potential” in referring to cryo-EM “maps” to prevent confusion with molecular surface-mapped electrostatic (Coulombic) potential calculated using atomic coordinates, which we also present in this paper. We now use “cryo-EM map” throughout.

2) Line 80 “has been” suggested a previous study- use “was” to indicate this study.

We are not sure about this... We thought that a “had been” would indicate a previous study. A “has been” is something that has just happened. We also thought that “was” precedes “has been”. In any case, thank you for bringing this to our attention. We will discuss this with the proofs editor if the paper is accepted.

3) Lines 148-149. Could the insertion domain serve a similar role as sigma region 1 (prevent nonspecific DNA binding)? If it's negatively charged, why do the authors assume it interacts with DNA- I would think the opposite?

Thank you for your suggestion. We adjusted the text accordingly (top of page 8). Also of note, electrostatic repulsion is a bona fide interaction.

4) Lines 190-191: region 4 is just as conserved structurally as those sigmaD2 and finger, so the statement is incorrect.

Sequence-wise, region 2 is better conserved (Suppl Fig. 4). We have changed the text to avoid confusion nevertheless.

5) Lines 240-241. I think the authors should mention that the interactions likely facilitate stabilization of bubble and unwinding (not just DNA recognition).

Thank you for this suggestion. Your suggestion has been incorporated into the new version of the MS (bottom of page 12).

6) Figures can have better labels (Indicate in the figures which are bacterial, AR9, Xray, or Cryo-EM) in figures 2-8.

The labels have been added to the figure panels and/or figure legends.

REVIEWERS' COMMENTS

Reviewer #3 (Remarks to the Author):

The authors have done an excellent job addressing my questions/concerns. I recommend publishing this excellent manuscript.